



# Wind turbine main-bearing lubrication - Part 2: Simulation based results for a double-row spherical roller main-bearing in a 1.5 MW wind turbine

Edward Hart[1], Elisha de Mello[2], and Rob Dwyer-Joyce[2]

[1]Wind Energy and Control Centre, Department of Electronic and Electrical Engineering, The University of Strathclyde, Glasgow, UK
[2]Leonardo Centre for Tribology, Department of Mechanical Engineering, The University of Sheffield, UK

**Correspondence:** Edward Hart (edward.hart@strath.ac.uk)

**Abstract.** This paper is the second in a two-part study on lubrication in wind turbine main-bearings. Where "Part 1" provided an introductory review of elastohydrodynamic lubrication theory, this paper will apply those ideas to investigate lubrication in the double-row spherical roller main-bearing of a 1.5 MW wind turbine. Lubrication is investigated across a "contact conditions dataset" generated by inputting processed loads, obtained from aeroelastic simulations, into a Hertzian contact model of the main-bearing. From the Hertzian model is extracted values of roller load and contact patch dimensions, along with the time rate-of-change of contact patch dimensions. Also included in the dataset are additional environmental and operational variable values (e.g. wind speeds and shaft rotational speeds). A suitable formula for estimating film thickness within this particular bearing is then identified. Using lubricant properties of a commercially available wind turbine grease, specifically marketed for use in main-bearings, an analysis of film thickness across the generated dataset is undertaken. The analysis includes consideration of effects relating to starvation, grease thickener interactions and possible dynamic EHL effects. Results show that the modelled main-bearing would be expected to operate under mixed lubrication conditions for a non-negligible proportion of its operational life, indicating that further work is required to better understand lubrication in this context and implications for main-bearing damage and operational lifetimes. Key sensitivities and uncertainties within the analysis are discussed, along with recommendations for future work.

## 1 Introduction

Higher than expected failure rates for wind turbine main-bearings have led to increased research focus on this component in recent years (Hart et al., 2019, 2020; Guo et al., 2021). Main-bearing failures are costly, with their replacement generally requiring the entire turbine rotor to be removed and supported during changeovers. As turbines move further from shore, component reliability becomes increasingly important due to additional costs associated with heavy lifting vessel procurement and operation, access lead-times, and impacts of lost revenue on the levelised cost of energy. Furthermore, the mechanisms leading to premature main-bearing failures are still not properly understood (Hart et al., 2019; Guo et al., 2021). This gap in knowledge, regarding fundamental causal mechanisms, represents a risk to the reliability of newer (larger) wind turbines,





since it cannot currently be known *a priori* whether main-bearing failure rates are likely to be improved or worse for these, and future, machines. In addition, practical solutions to ameliorate the current rates of main-bearing failure cannot be system-
25 atically identified, tested and developed until principal failure drivers in the wind turbine context are better understood. As argued in Hart et al. (2020), an improved understanding requires the full load pathway - including the turbulent wind field and aerodynamic/control interactions which drive the loading which passes into the drivetrain and then the main-bearing - to be accounted for.

As outlined, the main-bearing performs the important task of supporting the rotor and reacting (to a greater or lesser extent)
non-torque loads entering the drivetrain; crucially, this must be achieved while providing low-friction free-rotation of the shaft and without rapid wear to bearing internal surfaces. Wind turbine main-bearings therefore contain a lubricant, the role of which is to fully or partially separate bearing internal surfaces, via fluid and elastic-solid interactions, greatly improving frictional conditions and minimising wear. The lubricant and lubrication mechanisms are therefore fundamental to main-bearing oper- ation and lifetime and, as a result, must be considered if internal main-bearing conditions and possible damage drivers are
to be properly investigated. Indeed, the fatigue life assessment process for rolling bearings explicitly assumes that elastohy- drodynamic lubrication (EHL) conditions hold throughout the bearing lifetime (Hart et al., 2020). The validity, or not, of this assumption in the main-bearing context therefore has direct and important implications for the design and life assessment of this component.

In order to begin addressing the problems outlined above, the present study investigates lubrication in a 1.5 MW wind turbine
main-bearing under realistic load and speed conditions obtained from wind turbine simulations. Lubrication is considered by applying film thickness equations along with other results from EHL theory. The fields of lubrication and EHL are complex, nuanced and rapidly evolving. As such, simplified lubrication equations and associated results must not be applied blindly, but with careful consideration of their validity and context. To this end, "Part 1" of the present study (Hart et al., 2021) is a detailed introductory review of EHL theory which seeks to provide an accessible and informative overview of the field.
This manuscript, "Part 2", details the main-bearing lubrication analysis itself, including: simulations, internal load evaluation, lubricating conditions and film-thickness analyses. To aid the reader, a table of symbols is provided in Appendix A.

## 2  Background

Main-bearing research to-date has mainly focused on load modelling (Kock et al., 2019; Hart, 2020; Wang et al., 2020; Zheng et al., 2020a; Stirling et al., 2021), load characteristics (Cardaun et al., 2019; Hart et al., 2019; Hart, 2020; Guo et al., 2021) and
50 implications for fatigue damage (Zheng et al., 2020b; Loriemi et al., 2021). Lubrication conditions within the main-bearing are generally not considered. Two exceptions to this are Rolink et al. (2020) and Guo et al. (2021). In Rolink et al. (2020), EHL films are modelled as part of a general feasibility study of a novel plain-bearing solution for the main-bearing in wind turbines. In Guo et al. (2021), axial motion of the shaft and main-bearing inner ring were investigated using both direct measurements and the application of simplified models. The main aim of the study was to ascertain whether axial motion in an operating main-
55 bearing is rapid enough to potentially disturb the EHL film. It was concluded that axial motions are slow and highly unlikely to



impact the lubricating film in these components. This conclusion is valuable, in that it helps narrow down the possible causes of premature failures in main-bearings. Note, lubrication was not modelled directly in this previous study. Therefore, to the best of the authors' knowledge, a detailed analysis of lubrication conditions and film thickness in a wind turbine main-bearing, under realistic operating conditions, has not been presented before in the scientific literature.

Some previously undertaken modelling work was again used in this study. This relevant literature will therefore be described in more detail. Particularly relevant to the current study is Hart (2020), in which large repeating load patterns were shown to be experienced by the main-bearing throughout operation. In addition, a Hertzian contact model of a double-row spherical roller main-bearing was presented and used to investigate individual roller loads during operation. The described load structures were found to drive large, rapid variations in individual roller loads inside of the main-bearing. The same Hertzian model,
of a double-row spherical roller main-bearing, is again used in the current investigation. It is important to note that the axis labels ($x$ and $y$) assigned to rolling and transverse directions in Hart (2020) differ from those generally used in EHL modelling. Throughout this present study the standard EHL direction pairings are used ($x$ = rolling direction, $y$ = transverse to rolling), see Hart et al. (2021). Load inputs to the Hertzian model are obtained using a simplified drivetrain representation and a static load balance, between applied hub loading and main-bearing response, at each time-step (Hart et al., 2019). Applied loads at the
turbine hub are obtained from simulations (described below) and the radial loads at the main-bearing are resolved independently in horizontal and vertical planes. In Stirling et al. (2021) it was shown that, for the single main-bearing configuration, this simplified drivetrain representation is able to accurately recreate the radial response at the main-bearing seen in a higher fidelity, but still relatively simple, 3D finite element model. All thrust is assumed to be reacted by the main-bearing for the single main-bearing configuration (equivalently, "three point suspension") studied here. The total load resulting at the main-
bearing, which forms the input to the Hertzian contact model, is a vector containing the two radial and one axial (thrust) components.

Also relevant are the findings of Kock et al. (2019), who showed that elastic effects beyond those of just the contacts themselves can play an important role in determining main-bearing internal loading. More specifically, it was shown that neglecting influences of elastic surroundings and bed-plate flexibility can lead to roller load over-prediction of up to about 50%.
However, their analysis was undertaken for the downwind cylindrical roller bearing of a double main-bearing (equivalently, "4 point suspension") configuration, within a large (6MW) wind turbine. A number of differences are therefore present between their 6MW turbine analysis and the 1.5MW turbine (3 point suspension) drivetrain considered here, with size being the major one, since flexibility would be expected to play a greater role as the modelled turbine and its components become larger. The Hertzian model of Hart (2020), which was used in the current study, does not account for flexibility in the bearing system other
than that at roller-raceway interfaces. As such, it may be over-predicting individual roller loads to some extent, as described in Kock et al. (2019). But, since our Hertzian model is for a considerably smaller bearing, load over-prediction would be expected to be less severe than the 50% reported for the larger bearing model. The level of possible over-prediction is unknown, therefore, a load sensitivity analysis for lubrication results will be presented to address this.

Wind turbine simulations in previous and current work were undertaken using DNV-GL Bladed software. Bladed is a design
certified wind turbine aero-elastic simulation tool. Aerodynamic interactions are evaluated using a blade-element-momentum




implementation, with structural response evaluated via a multibody formulation of the wind turbine's structural dynamics. Turbine loads generated using Bladed include the effects of aerodynamic, elastic, inertial and gravity driven loading. Tilt is also present for the modelled turbine rotor, effects of which are automatically accounted for in existing drivetrain models since the hub frame of reference (and so also the force/moment outputs) is aligned with the tilted low-speed shaft. Nonsteady wind

fields with which simulated turbines interact are generated by combining deterministic components (such as tower-shadow, shear etc.) with kinematic turbulence, generated using a specified spectral model (Hart et al., 2020).

A thorough treatment of EHL theory has been given in Part 1 of this study (Hart et al., 2021). As such, familiarity with the material presented there is assumed throughout this paper.

## 3  Methodology

In order to study lubrication in the case of a wind turbine main-bearing, an extensive "contact conditions dataset", including roller loads, shaft speeds and contact dimensions (along with other relevant quantities), was generated. These values, combined with lubricant data and additional bearing information, allow film estimation formulas and EHL results to be applied (Hart et al., 2021). The process by which this dataset was generated will be detailed first, followed by a summary of bearing and lubricant information relevant to the study. The methodology for the lubrication analysis itself will then be described.

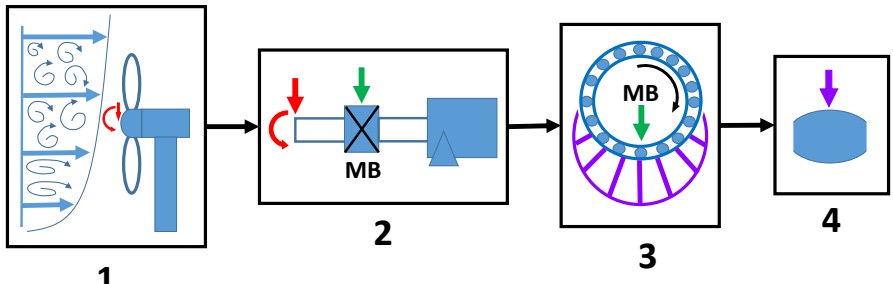

**Figure 1.** Visual summary of the process by which the "contact conditions dataset" was generated. Starting from turbulent wind interactions, loads introduced to the system are resolved in the drivetrain and then inside of the main-bearing. The resulting dataset captures the load acting at individual rollers within the main-bearing, along with other information important to lubrication.

## 105  3.1  Generating the "contact conditions dataset"

Figure 1 summarises the process by which contact conditions within the main-bearing were evaluated. Background on the software and models used was provided in Section 2. Dataset generation followed a four stage process:

1. Aeroelastic simulations of a 1.5 MW (variable speed, pitch regulated) wind turbine across a range of hub-height mean wind speeds (6-24 m/s, in increments of 2 m/s) and turbulence levels (low, medium and high as defined by the IEC design

standards) were performed using DNV-GL Bladed software. From each (10-min) simulation was extracted 100Hz time-





series' of hub loading in six degrees of freedom (3 force components and 3 moment components). Environmental and operational time-series (shaft speed, rotor mean wind speed, turbine power, blade pitch angle etc.) were also extracted.

2. Applied radial loading at the main-bearing was then evaluated, taking the hub-loads from the previous step as input. This was done using the statically determinate single main-bearing model previously used in Hart et al. (2019); Hart (2020); Stirling et al. (2021) and discussed in Section 2. As described earlier, evaluation of main-bearing applied loading involves a static load balance at each timestep. Note, as was the case in Hart (2020), the required load for inputting to the Hertzian model is that being *applied* to the bearing by the shaft, as opposed to the *reaction* of the bearing. This difference is subtle, since the two forces have equal magnitude but opposite direction. It is necessary to ensure the correct one is calculated with respect to the reference frame used. All thrust loads are assumed to be reacted by the main-bearing.

3. Load conditions within the main-bearing, resulting from applied loads, were then estimated using the Hertzian contact model presented in Hart (2020), and for the same bearing. At each timestep, an optimisation is performed wherein radial and axial deflections of the main-bearing inner ring are found such that reaction forces from roller deflections balance the applied radial and thrust loads. This model is therefore quasi-static in nature and does not account for possible inertial effects within the bearing. The deflections, once solved for, fully characterise roller loading within the bearing.

4. Roller loads were then extracted for each roller in the downwind row of the double-row main-bearing. This is because the upwind row is only occasionally loaded during normal operation, as a result of thrust (Hart, 2020). From roller load was also calculated the Hertzian semi-widths ($a$ and $b$) of the resulting contact patch at inner and outer raceway contacts (Hart, 2020). By evaluating the system at the next timestep, finite difference was used to estimate the rate of change of these quantities in time, *i.e.* $da/dt$ and $db/dt$. These values are important for determining whether dynamic EHL effects may be significant here (Hart et al., 2021). Roller trajectories from one timestep to the next were evaluated assuming pure rolling. Roller loads, contact patch semi-widths and semi-width time derivates were all included in the resulting dataset, along with environmental and operational values corresponding to the same point in time.

To avoid data quantities becoming unwieldy, data was extracted for each loaded roller in the downwind race at 200 randomly selected points in time from within each 10-min simulation. The resulting contact conditions dataset, from a total of 30 simulations in varying conditions, contains over 126,000 roller load (and associated operating conditions) entries from across the turbine's operational envelope.

## 3.2 Main-bearing and lubricant data

As described in Hart et al. (2021), lubrication analysis requires information relating to bearing geometry, surface roughness and lubricant properties. The modelled bearing is an SKF 240/630CA/W33, which is used for wind turbines of the size being simulated in this work. Some of its geometric information is available in the public domain, however, other details are proprietary and so cannot be given explicitly. The lubricant which will be modelled in the current analysis is an industrial grease, specifically designed and marketed as a lubricant for wind turbines of this size, including for main-bearings. Table 1



**Table 1.** Main-bearing and lubricant data

| | |
|---|---|
| Bearing | SKF 240/630CA/W33 |
| Combined surface-roughness of roller-raceway interfaces | $\sigma \approx 300$ nm |
| Inner- and outer-raceway reduced radii in rolling direction | $R_x^{\mathrm{in}} \approx 0.03$ m, $R_x^{\mathrm{out}} \approx 0.04$ m |
| Inner- and outer-raceway contact ellipticity parameter | $k^{\mathrm{in}} \approx 42$, $k^{\mathrm{out}} \approx 38$ |
| Reduced elastic modulus $\left(2/E' = (1-\nu_{\mathrm{I}}^2)/E_{\mathrm{I}} + (1-\nu_{\mathrm{II}}^2)/E_{\mathrm{II}},\text{ see Appendix A}\right)$ | $E' = 225.3$ GPa |
| Inlet dynamic viscosity (at atm. pressure) at 35 $^\circ C$ | $\mu_o = 0.6525$ Pa·s |
| Inverse asymptotic isoviscous pressure (assumed value) | $\alpha^* = 21$ GPa$^{-1}$ |

presents information relating to this bearing and lubricant, approximate values are given where necessary. Kinematic viscosity information, from the grease manufacturer, was provided at 40 $^\circ C$ (460 mm$^2$/s) and 100 $^\circ C$ (16 mm$^2$/s) along with the
145 relative density (0.9), as is standard. Interpolation to other temperatures was performed using the ASTM prescribed method (ASTM, 2020). The inverse asymptotic isoviscous pressure, $\alpha^*$, of the main-bearing grease base-oil is not known. In Vergne and Bair (2014) a range of measured $\alpha^*$ values are listed. At a temperature of 40°C, reported measured values have a mean value plus/minus standard deviation range of $\alpha^* = 21 \pm 6$ GPa$^{-1}$. While $\alpha^*$ is temperature dependent, it will not vary much for small differences in temperature (on the order applied here, see Vergne and Bair (2014)). A value of $\alpha^* = 21$ GPa$^{-1}$ is
150 therefore assumed, this being the best available estimate without more information. The sensitivity of results to the $\alpha^*$ value is also considered as part of the analysis.

### 3.2.1 Operating temperatures for the main-bearing

Operating temperatures will strongly influence lubricant viscosity, which in turn has a significant effect on lubricant film thickness. The simulation software and models used to evaluate contact conditions in the current work do not model or predict
operating temperatures. It is therefore necessary to identify an appropriate temperature range over which to consider main-bearing lubrication. Based on published temperature data for main-bearings in the literature (de Mello et al., 2021; Beretta et al., 2021), normal operation of healthy main-bearings can be seen to include temperatures of around 20-40 $^\circ C$ as standard, with higher temperatures seen if a fault develops. Temperature values reported in the literature relate to measurements taken on the outside of the bearing casing, as opposed to inside of the main-bearing. Lubricant inlet temperatures may therefore be slightly
higher than the values listed above. Preliminary analyses, during the course of this work, revealed that an important transition in lubrication regime occurs at around 35 $^\circ C$ for the modelled main-bearing and lubricant. Therefore, the analysis presented here will be centred on this value, with results for higher and lower temperature cases of 40 $^\circ C$ and 30 $^\circ C$, respectively, also given. It is again emphasised that, based on published data, explored temperatures fall within the standard operating temperature range of healthy main-bearings. For analysis at a given temperature, that same temperature is assumed to hold across all operating
points in the contact conditions dataset. Temperatures in the main-bearing will vary during operation as a result of frictional





effects and internal load and speed interactions. However, significant thermal inertia is present (de Mello et al., 2021), meaning only relatively weak correlations exists between load, speed and temperature measurements. As such, each operating point may be seen at a range of operating temperatures, indicating that the application of lubrication equations across all operating points while using a single fixed temperature does provide a valid assessment of conditions that will be seen in practise.

### 3.3 Lubrication analysis methodology

While highly sophisticated EHL solvers have been developed, as has been outlined (Hart et al., 2021) setting up and running these solvers is non-trivial. Prior to the application of such methods, it is sensible to first scrutinise the problem using simplified lubrication equations and other results from EHL theory. This provides immediate insight into main-bearing lubrication conditions and behaviour, allows the need for more advanced solvers in this space to be assessed, and, helps determine where more sophisticated investigations might be best focused. Therefore, in order to estimate surface separation within the main-bearing, for the conditions identified using the outlined modelling approach, an appropriate film thickness equation must be identified. In the present case, this includes determining whether roller-raceway contacts should be treated as line- or point-contact conjunctions. From Table 1, it can be seen that very high ellipticity values hold for contacts within the main-bearing in question. In addition, the relative magnitudes of required adjustments to inner- and outer-raceway elasticity values when forming equivalent line contact representations (Hart et al., 2021) were found to be negligible (around 0.1%) for the modelled bearing. This indicates that roller-raceway contacts in this bearing are very close indeed to their equivalent line-contact counterparts to begin with. Furthermore, the ellipticity values themselves far exceed those for which point-contact film thickness equations have been developed (Hart et al., 2021). It was therefore concluded that film thickness for this bearing should be analysed using the equivalent line contact formulation, outlined in Hart et al. (2021), in combination with a line-contact film thickness formula. The formula selected for this analysis was the more recent and comprehensively fitted Masjedi and Khonsari line contact equation (Hart et al., 2021; Masjedi and Khonsari, 2012). Given the dimensionless roughness values of the main-bearing bearing in questions, $\sigma/R_x^{\text{in}} \approx 9.7 \times 10^{-6}$ and $\sigma/R_x^{\text{out}} \approx 8.1 \times 10^{-6}$, it follows that the associated rough-surface correction factor should also be used (Hart et al., 2021). Therefore, minimum film thickness was calculated as,

$$
\begin{aligned}
h_m &= 1.652 \frac{U^{0.716} G^{0.695}}{W_l^{0.077}} \Phi_l R_x \\
&= 1.652 \frac{U^{0.716} G^{0.695}}{W_l^{0.077}} \left( 1 + 0.026 \left( \frac{s_{\text{std}}}{R_x} \right)^{1.120} V^{0.185} W_l^{-0.312} U^{-0.809} G^{-0.977} \right) R_x.
\end{aligned}
\tag{1}
$$

For definitions of all terms see Appendix A and Hart et al. (2021). Assuming normally distributed roughness it follows that $s_{\text{std}} = \sigma$. Bearings of this type are hardened as much as possible and so a value of $V = 0.03$, corresponding to a high Vickers hardness of 700 HV which is typical for bearing steel, is used throughout. Lubricant entrainment velocity, $\tilde{u}$, is calculated assuming pure rolling, see Appendix B. Prior to the lubricating film results, contact pressures and dimensionless lubrication parameter values for the main-bearing will be presented. As will be shown, operating parameter values for the main-bearing lie well within the limits of where Equation 1 was fitted.





Significant uncertainty is present regarding conditions inside of the main-bearing which contribute to overall lubrication performance, especially due to the likely presence of some level of starvation under grease lubrication (after the initial churning phase) (Hart et al., 2021). Lubrication was therefore investigated starting from a "best case scenario" of fully-flooded conditions, for which Equation 1 is applied directly while assuming lubrication is driven by the grease base-oil properties. Next, obtained film thicknesses were reduced to 71% of their fully-flooded values, to represent starvation at a level reported for conditions of zero-reverse-flow and also consistent with the reduction proposed for cases of starved grease lubrication made in a previous study (Hart et al., 2021). As discussed in Part 1, this offers only a crude estimate of starvation effects which, in practise, will vary with speed and other parameters. For grease lubrication in particular, considerably more severe levels of starvation have been observed (Cen and Lugt, 2019, 2020). Both fully-flooded and starved results will be presented for temperatures of $T = 30,\ 35,\ 40°C$, allowing the sensitivity of film thickness to operating temperature to be quantified. Roller load and $\alpha^*$ value sensitivities were also investigated. The approach outlined above does not account for possible grease thickener interactions on film thickness values. The importance of this aspect of grease lubrication in the main-bearing case will therefore be considered by combining film thickness results, obtained here, with results reported in the literature concerning thickener behaviour. The analyses outlined thus far all rely on the chosen film thickness equation, which applies to steady-state lubrication. Possible dynamic effects are therefore not accounted for, with the above therefore referred to as *steady-state film thickness analyses*. The possible presence of significant dynamic EHL effects during main-bearing operation is investigated in a subsequent analysis, by considering $db/dt$ and $da/dt$ values relative to the concurrent value of lubricant entrainment velocity $\tilde{u}$. A lower limit for significance is taken to be 25% based on findings reported in the literature (Hart et al., 2021).

At this stage, it is not possible to quantify the accuracy with which the described lubrication analysis is able to represent lubrication conditions, behaviour and film thickness in a real world main-bearing. Results must therefore be interpreted with care, keeping in mind the related discussions in Part 1 of this work (Hart et al., 2021). However, this analysis should allow for general conclusions to be reached regarding the dominant lubrication regime(s), key film thickness sensitivities and the likely importance of more complex effects (thickener interactions/dynamic effects). This, therefore, is the context in which results should be approached and interpreted.

## 4 Results

Lubrication analysis results will be presented in the current section. It is first important to revisit the approximations made during modelling stages. This includes the half-space approximation for contacting surfaces (in Hertzian and EHL models used to generate film thickness equations), as well as the lubrication approximation applied in the same EHL models. In addition, the outlined methodology analyses bearing internal loading and displacement entirely independently of lubrication and lubricant film thickness values. Such an approximation is only valid if one does not have a significant impact on the other, more specifically, if roller deflections are significantly larger than film thickness values. Assessing the appropriateness of these approximations, in the current case, therefore requires model geometries and outputs to be compared. Such an analysis was





undertaken, confirming (as far as is possible without more complex analyses) that all requirements are met with regards to ensuring modelling approximations may be considered valid. The analysis itself is presented in Appendix C.

## 4.1 Lubrication conditions

Prior to film analysis itself, it is necessary to consider the conditions (with respect to dimensionless parameter values) under which lubricated conjunctions within the main-bearing are operating. This allows for the extremity of conditions to be assessed, while also providing an indication of the validity of a chosen film thickness equation. As outlined in Section 3.3, the main-bearing analysis was conducted while treating the point contact conjunctions as equivalent line-contacts. Dimensionless parameters in this case are therefore those of the equivalent line contact. Moes' load and viscosity parameter values ($M_l$ and $L$) were calculated across the contact conditions dataset at each investigated temperature. Figure 2a shows the dimensionless parameter results at 35°C, along with dimensionless parameter "fitting region limits" for the selected film thickness equation (see Hart et al. (2021)). Both inner- and outer-raceway parameters are plotted, with considerable overlap occurring between the two. Points in the plot are coloured according to the rotor mean wind speed occurring at that point in time.

The range of viscosity parameter ($L$) values visited during main-bearing operation can be seen to be small. In particular, the values of $L$ at inner- and outer- raceway contacts correlates strongly with wind speed. This relationship stems from the turbine operating strategy, in which rotational speed increases with wind speed (until nearing rated power) in order to maintain aerodynamic efficiency (Hart et al., 2020). Variations in load parameter values ($M_l$) are considerably higher, with $M_l$ uncorrelated to wind speed. Both observations are unsurprising, given that significant load variation around the bearing circumference will be present (Hart, 2020) - generally including an unloaded region. Each roller traversing the bearing circumference therefore experiences continuous variations in loading from minimum (possibly zero) to maximum (in terms of roller loads around the circumference at a given point in time) levels. Thus, observing a wide range of dimensionless load parameter values at each wind speed would be expected. With respect to the fitting region limits of the chosen film thickness equation: dimensionless viscosity parameter ($L$) values can be seen to lie well within the range over which the equation was developed; dimensionless load parameter values ($M_l$) also lie within their associated limits for the vast majority of points in the contact conditions dataset, indeed, fitting region limits are only passed in the low-load region, as $w_l \to 0$. It should be noted that the outlying points at very low-loads correspond to high values of film thickness. With regards to steady-state lubrication analysis (via film thickness equations), it is regions of potentially low film thickness that are of primary interest; as such, it is at higher load levels that film thickness values are most important. As may be seen in the figure, all higher values of the dimensionless load parameter fall well within the limits of the applied film equation. To avoid erroneous results from extremely small load-parameter values (the smallest of which are of order $10^{-5}$), all results presented in subsequent sections exclude cases where $w < 1\text{kN}$. Given the operating speeds of the modelled turbine, this limit ensures $M_l \geq 1.4$, with all presented results therefore for parameter values falling within or close to the boundaries shown in Figure 2a. In Hart et al. (2021), literature misrepresentations of the parameter ranges over which the Masjedi and Khonsari point contact equation was fitted were described. It is pertinent to note that with regards to point contact dimensionless parameters ($L$ and $M$), operating points in the current dataset also fall well within the point contact fitting region limits at higher load levels, but, this appears not to be the case if the incorrectly reported limits are



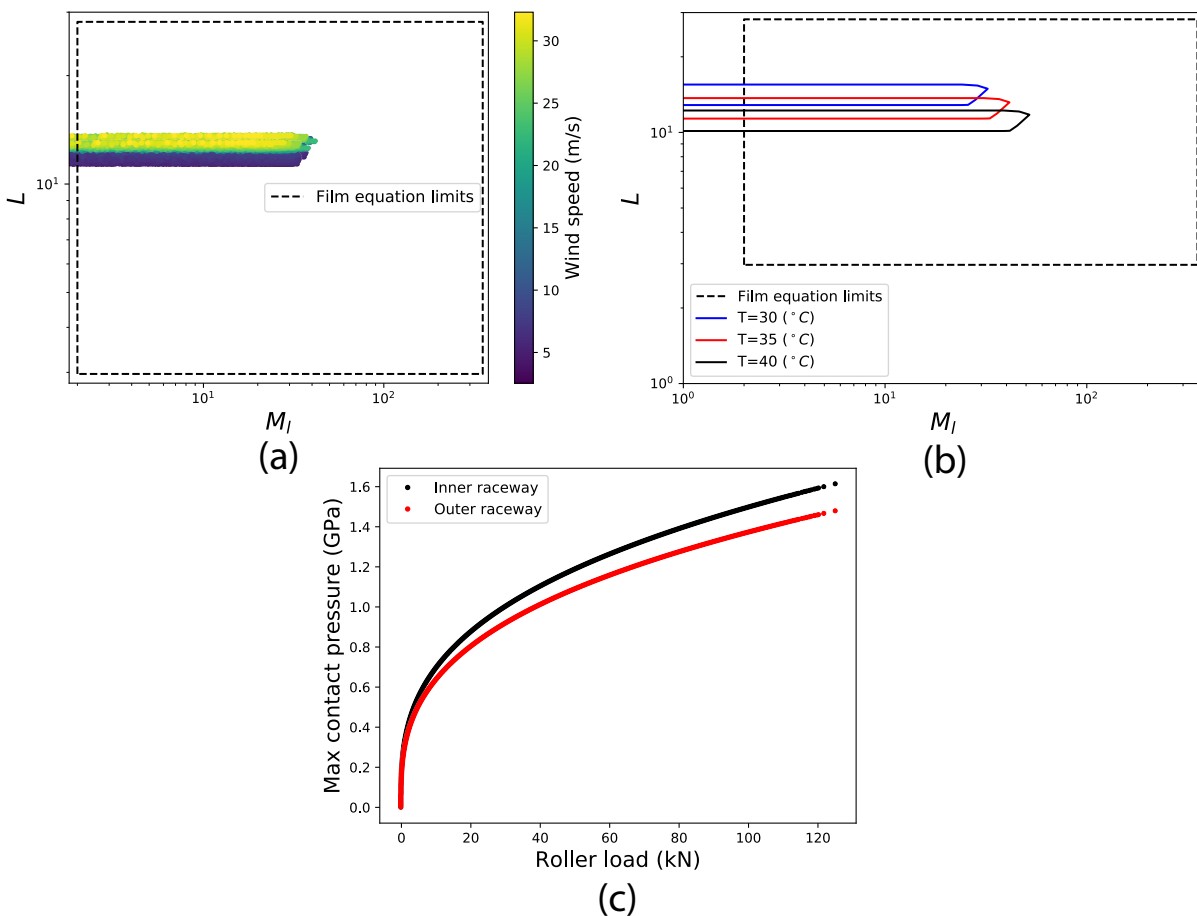

**Figure 2.** Lubrication conditions, in the form of dimensionless parameter values and maximum Hertzian pressures within contact conjunctions. Subplot a) shows dimensionless parameter results at $35°C$ for both inner and outer raceways, subplot b) the contact condition point boundaries for temperatures of 30, 35 and $40°C$, and subplot c) the maximum contact pressures from across the dataset.

used. While the point contact equation is not applied here, for reasons which have been outlined, this example demonstrates the need for fitting region limits to be checked, while also ensuring that the correct limits are applied.

Lubricant inlet temperature is known to be a strong driver of viscosity and, hence, of dimensionless parameter values and film thickness. The impact of temperature variations on dimensionless parameter values was therefore considered. Figure 2b shows boundaries of operating point sets (more specifically their convex hulls) obtained from the contact conditions dataset with temperatures of $T = 30$, 35 and $40°C$. Impacts of temperature on $M_l$ and $L$ values can be seen to be relatively small, with the previously made observations remaining valid in each case.

Finally, contact pressures within the main-bearing contact conjunctions were considered by approximating maximum EHL pressure values as those seen in their equivalent dry Hertzian contacts (see Hart et al. (2021)). By construction, the maximum





pressure within point and equivalent line contact representations of the conjunctions are equal (Hart et al., 2021). For inner- and outer-raceway contacts, Figure 2c shows maximum contact pressure values, plotted against roller applied loads, for all points in the contact conditions dataset. Surprisingly, these values are lower than might be expected given the high magnitude
of loads reacted by the main-bearing (Hart, 2020). For bearings of this type, the rated load level will sit at around 2.5-3 GPa. As shown in the figure, maximum contact pressures, obtained from the modelling undertaken here, all lie comfortably below these limiting values. The relatively modest pressure levels seen here are likely due to the combined effects of high curvature at contact interfaces (indicating that a larger surface-area will be actively engaged in supporting applied loads) and a relatively large number of rollers, 27, in each row. More complex internal effects such as roller unseating and skewing are not modelled
here, therefore, higher levels of pressure could occur in practise as a result of such interactions.

The presented lubrication conditions results indicate that, having applied the restriction $w > 1\text{kN}$, operating points fall such that the chosen film thickness equation is appropriate for performing a steady-state analysis of film thickness for this main- bearing. Note, the standard caveats regarding interpretation of results obtained from simplified film thickness equations (as discussed in Hart et al. (2021)) still apply. Contact pressure results indicate that, with respect to effects modelled currently, the
bearing material ultimate strength limits are not being exceeded during operation.

### 4.2   Steady-state film thickness analyses

Having identified a suitable low load cut-off and confirmed suitability of the chosen equation, steady-state film thickness analyses were undertaken as described in Section 3. The results of these analyses will now be presented.

**Fully-flooded results**: Figure 3a shows film parameter (equivalently the dimensionless film thickness), $\Lambda$, values obtained for
operating points in the contact conditions dataset with $T = 35°\text{C}$. Shaft speed and roller load values are also plotted. Note, shaft speed is proportional to the lubricant mean entrainment velocity, $\tilde{u}$ (see Appendix B). Inner raceway points are plotted individually whereas, for the sake of clarity, the overlapping outer raceway results are summarised via their convex hull. Focusing on inner raceway results, $\Lambda$ values can be seen to fall between 2.7 and 5.1, with a mean value of 3.8. As would be expected from the exponents in film equations, as well as the results of EHL theory more generally (Hart et al., 2021), film
thickness values are strongly driven by rotational speed at all load levels. Variations in film thickness with load also show a relatively wide spread, but, closer inspection reveals that most of this variation occurs at low roller loads, with film thickness load sensitivity falling dramatically as roller load values increase. This too is expected from general results of EHL theory. Note also that the smallest loads are generally experienced as rollers move towards the edge of the current load zone (Hart, 2020), which itself may move, while traversing the bearing circumference. Each roller will therefore move through a wide range
of load levels/$\Lambda$ values during each orbit. Outer raceway results are qualitatively similar, but offset by a small amount in the positive $\Lambda$ direction relative to those of the inner raceway. Since pure rolling is assumed, $\tilde{u}$ values are identical at inner and outer raceways (see Appendix B). Therefore, observed film thickness differences result from differing geometries at the respective contacts. With respect to the lubrication regimes associated with $\Lambda$ values (Hart et al., 2021), for fully-flooded operation at $T = 35°\text{C}$, both inner and outer raceway contacts are predicted to operate mainly in the elastohydrodynamic regime ($\Lambda > 3$),



with some small amount of time spent in the mixed lubrication regime ($1 < \Lambda < 3$)[1]. From Figure 3a, it is evident that the key factor (under fully-flooded conditions) determining the lubrication regime, and possible transition to mixed lubrication, is the shaft speed, with the least favourable conditions occurring at low speeds.

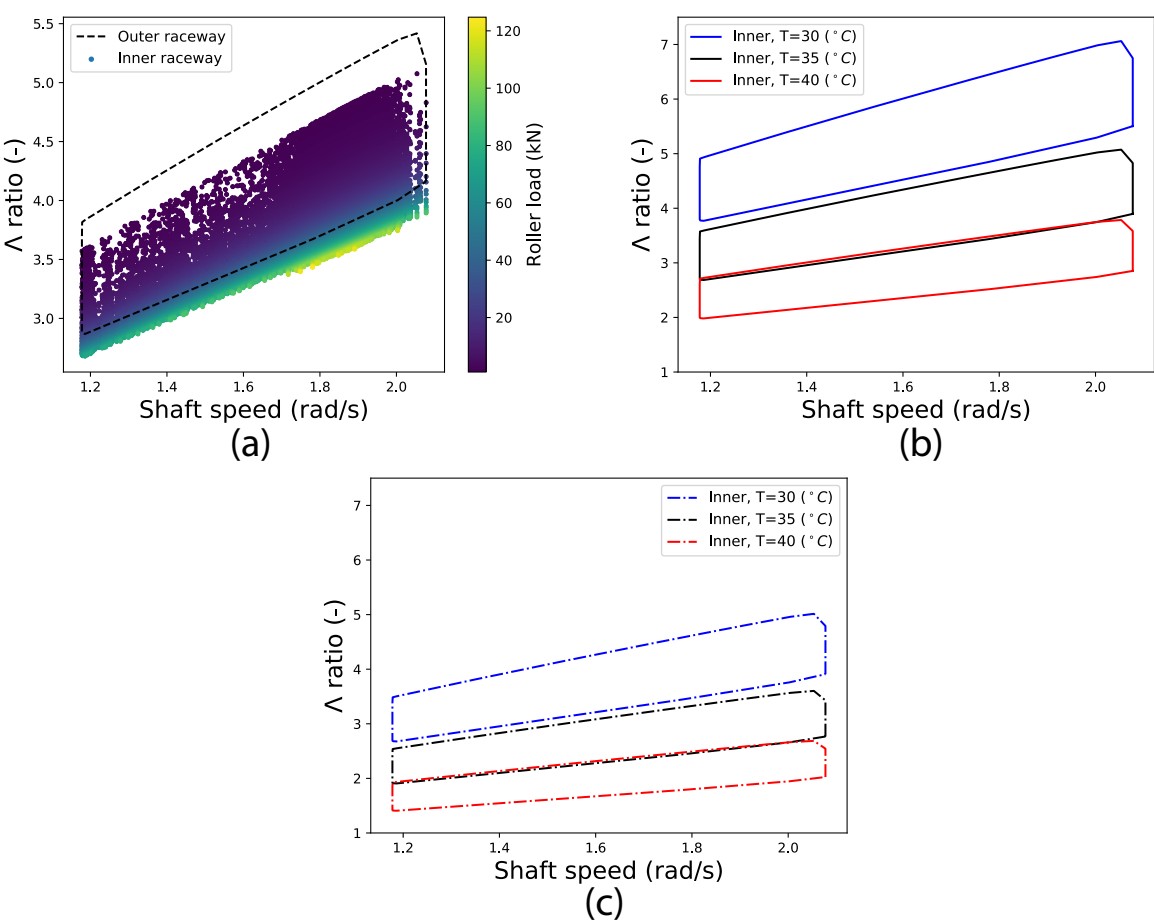

**Figure 3.** Steady-state dimensionless film thickness ($\Lambda$) results. Subplot a) shows inner and outer raceway results at $T = 35°C$, subplot b) shows inner raceway results when $T$ is varied and c) shows the same results having estimated the effects of starvation.

Figure 3b shows the effect of temperature variations on film thickness, with inner raceway $\Lambda$ value boundaries plotted from results obtained at temperatures of $T = 30, 35, 40°C$. As would be expected, from lubricant viscosity behaviour, a reduction in

operating temperature leads to an increase in film thickness, while reduced film thickness values are seen when the temperature

---

[1]There is some level of overlap between the lubrication regime designations relating to given ranges in $\Lambda$ (Hamrock et al., 2004; Hart et al., 2021). For the sake of simplicity, in the current analysis the standard ranges have been interpreted in their most optimistic light. Therefore, the elastohydrodynamic regime is taken to be $3 < \Lambda < 5$ (with hydrodynamic lubrication assumed to take place for $\Lambda > 5$) and the mixed regime taken as $1 < \Lambda < 3$. Boundary lubrication occurs for $\Lambda < 1$.





is increased. From the point of view of the lubrication regime, impacts of these relatively modest changes in temperature are dramatic. A reduction of $5°C$ can be seen to draw results well into the elastohydrodynamic regime and even into possible hydrodynamic lubrication. On the other hand, an increase of $5°C$ draws results significantly into the mixed lubrication regime, with 79% of operating points having $\Lambda < 3$. With respect to fully-flooded results, the operating temperature $T = 35°C$ therefore

represents an important transition point (for the modelled main-bearing and lubricant) in terms of operational lubrication regimes.

**Starved results**: Main-bearings are grease lubricated and, therefore, expected to be operating under starved conditions throughout most of their operational lifetimes (Hart et al., 2021). As detailed in Section 3.3, a crude estimate of the impact of starvation on main-bearing lubrication was obtained by taking 71% of the fully-flooded $\Lambda$ values. The effect of this is shown in Figure 3c.

As with temperature variations, the impacts here are significant, with all operating points at $T = 40°C$ now well into the mixed lubrication regime and beginning to near boundary lubrication ($\Lambda < 1$). For results at $T = 35°C$, 88% of operating points are now in the mixed regime. Starvation levels in operational main-bearings are not known and, in reality, will vary with the operating conditions. However, much higher levels of grease starvation than that applied here have been observed in practise (Hart et al., 2021). Therefore, even having applied a reduction factor, the results in Figure 3c could still be providing an optimistic

view of main-bearing lubrication.

**Grease thickener interactions**: As described in Part 1 of this study (Hart et al., 2021), at film thicknesses below a level related to the size of thickener fibres/fibre-networks the thickener interacts with the contact conjunction, altering film thickness behaviour such that it can no longer be estimated using oil lubrication formulas with properties of the grease base-oil. Determining if such effects may be present for the modelled main-bearing therefore requires estimated film thickness values to be compared

with a suitable "transition thickness", above which base oil effects would dominate and below which the film behaviour would be driven by thickener interactions. Such a value is unknown for the main-bearing grease modelled here. Instead, results reported in the literature are used to estimate a sensible range for such a transition thickness. In Cen et al. (2014) and Kanazawa et al. (2017), grease thickener interactions are investigated for a number of different greases, in the former, the impact of the grease being mechanically worked is also assessed. From results presented in these previous studies, maximum observed

values of transition film thicknesses sit at around 100nm. Transition values, along with the relative magnitude of thickener effects, reduce as the grease is worked, with worked grease transition thicknesses of around just 10nm seen in some cases. It is therefore proposed that a reasonable range in which to assume that transition thickness lies, for the main-bearing grease, is between 10 and 100nm. From Table 1 $\sigma \approx 300$nm, and so Figure 3 results (in terms of $\Lambda$) can be converted to approximate film thickness values using $h \approx 300 \cdot \Lambda$ nm. Therefore, $\Lambda = 3$ corresponds to a film thickness of $h \approx 900$nm and $\Lambda = 1$ corresponds

to $h \approx 300$nm. Across all results presented here, as well for more severe cases of starvation than was considered, film thickness values remain significantly higher than the maximum value in our approximate range for transition film thickness. Based on the available information a tentative conclusion is therefore reached, this being that it currently appears unlikely that grease thickener interactions significantly impact lubrication in the modelled main-bearing. This result also implies that, at present, there is no reason to assume that oil-based film equations, such as the one applied here, are not able to provide sensible estimates of



lubrication behaviour in this setting. The normal caveats regarding accuracy of such equations remain (see Hart et al. (2021)). It
is emphasised that these conclusions are tentative. For example, if fibre/fibre-network dimensions in main-bearing greases are
significantly larger than those of greases used in the cited literature, a different conclusion might result. Furthermore, surface
roughness may also influence the point at which grease fibre interactions begin to influence lubrication behaviour. In Cen et al.
(2014) and Kanazawa et al. (2017) surface roughness values are such that $\Lambda$ ratios are consistently greater than 1, never falling

much below this value if at all. If starvation is really quite severe for the main-bearing, $\Lambda$ values less than 1 are certainly possi-
ble, if so, it is not clear how operation inside of the boundary regime might impact grease interactions. Finally, it is important
to note that the studies referred to both ensure fully-flooded inlets at all times. While the discussion has thus far centred mainly
on film thickness values/ratios, it may also be the case that starvation impacts the likelihood of grease fibre interactions in
other ways, for example by altering the quantity of thickener fibres available to the conjunction or influencing the deposition

of degraded grease onto bearing surfaces (Hart et al., 2021). Much uncertainty therefore remains with respect to this particular
aspect of the problem.

The presented steady-state results provide important insight into the lubrication problem for wind turbine main-bearings.
However, it should be remembered that certain approximations and assumptions are present in the applied model and chosen
lubricant properties. The sensitivity of results to variable values associated with these sources of uncertainty was therefore con-

sidered. Sensitivity, with respect to $\Lambda$ values, was assessed for the variables of temperature ($T$), inverse asymptotic isoviscous
pressure ($\alpha^*$) and load ($w$). Temperature was considered as part of the above analysis, where it was found that its effect is
significant. It was therefore concluded that the impact of temperature variations should be quantified here along with the other
variables of interest. The true $\alpha^*$ value for the modelled lubricant is unknown and so an assumed value was used, the sensitivity
to changes in this variable will indicate how critical it is to ensure an accurate value in known. Finally, as outlined in Section 2,

results in the literature indicate that Hertzian models of the type applied here may result in over-estimations of the load levels
around the bearing circumference. In order to determine possible impacts of this to lubrication results, the effect of reducing
load levels was tested. The "standard case" was taken as being $\Lambda$ values calculated for $T = 35\,°C$, $\alpha^* = 21\,GPa^{-1}$ and with
load values unaltered. Each variable was then adjusted independently to determine its effect on $\Lambda$ values. For temperature,
variations of $\pm 5\,°C$ were applied; for $\alpha^*$, variations were $\pm 6\,GPa^{-1}$ (see Section 3.2); for $w$ the effect of all roller loads being

halved was determined. Graphical results for temperature variations have been shown above, those for load and $\alpha^*$ variations
can be found in Appendix D. Table 2 summarises the results of the sensitivity analysis, showing the mean-percentage-change
to $\Lambda$ values (across the contact conditions dataset) resulting from the specified changes to each variable. Temperature is the

| Variable | Change in variable ($\tau$) | $\Delta\Lambda(+\tau)$ [mean%] | $\Delta\Lambda(-\tau)$ [mean%] |
|---|---|---|---|
| Temperature | 5°C | $-26\%$ | $+41\%$ |
| $\alpha^*$ | 6 GPa$^{-1}$ | $+18\%$ | $-19\%$ |
| Roller load | $w \to w/2$ | - | $+6\%$ |

**Table 2.** Summary results of the sensitivity analysis. Mean percentage changes to $\Lambda$ values, resulting from specified changes to each variable's "standard value", are given.





most sensitive of the tested variables and, in practise, will vary quite considerably during operation. $\alpha^*$ value variations are also impactful with respect to estimated film thickness values. Determining accurate/appropriate values for this parameter will therefore be important to future main-bearing lubrication studies. Interestingly, and as might be expected from classical EHL

theory, the results are insensitive to variations in loading. A 50% reduction in loads ellicits only a 6% increase in $\Lambda$ values. This indicates that concerns related to modelled internal loads (see Section 2) are unlikely to have much influence on the findings of this EHL study. That is not to say that load levels and load distributions are not relevant to main-bearing lubrication, a point which will be revisited in Section 5, below. Starvation should also be considered in the context of these sensitivity results since it has a relative impact on the order of 30+%, with the true value potentially quite high. Based on the above, key sensitivities

for main-bearing lubrication are concluded to be: temperature, starvation levels and the lubricant $\alpha^*$ value.

### 4.3 The significance of dynamic effects?

Considerations of possible dynamic effects will be undertaken treating the point contacts as such, as opposed to their equivalent line representations used in the context of film thickness estimation, as above. As described in Section 3.3, the possible presence of dynamic EHL effects was investigated by considering contact patch time rate-of-change values in rolling and transverse

directions ($db/dt$ and $da/dt$, respectively) relative to the concurrent value of mean entrainment velocity ($\tilde{u}$). The relevance threshold is taken to be 25%. Results at inner and outer raceways are almost identical, and so only inner raceway results are shown here. Figure 4 shows contact conditions dataset values of $\dot{b}/\tilde{u}$, where $\dot{b} = db/dt$, plotted against roller load. As

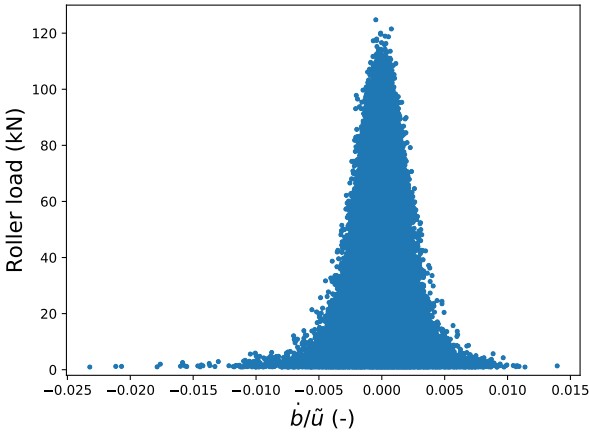

**Figure 4.** Inner raceway $\dot{b}/\tilde{u}$ (for $\dot{b}$ in the direction of rolling), where $\dot{b} = db/dt$, values plotted against roller load. Results are only shown where roller loads exceed 1kN.

previously, results are only shown where roller load exceeds 1kN. Considering these results, it is clear that $\dot{b}/\tilde{u}$ values fall well below the 0.25 which would signify the possible presence of significant dynamic EHL effects. Indeed, the estimated values are

390 at least an order of magnitude out from this. Rolling direction dynamic effects are therefore not expected to be occuring for the modelled main-bearing. Figure 5a shows contact conditions dataset values of $\dot{a}/\tilde{u}$, where $\dot{a} = da/dt$, plotted against roller


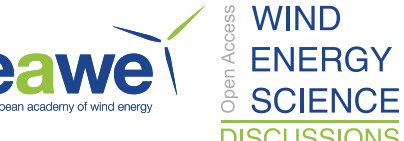


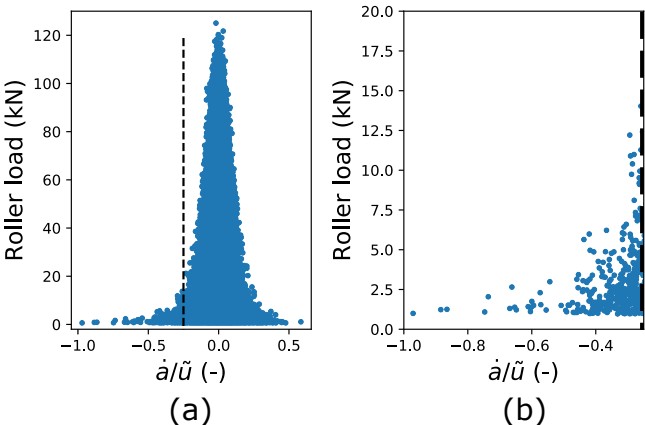

**Figure 5.** Inner raceway $\dot{a}/\tilde{u}$ (for $\dot{a}$ transverse to the direction of rolling), where $\dot{a} = da/dt$, values plotted against roller load. Results are only shown where roller loads exceed 1kN. Subplot a) shows all results, with a vertical dashed line included to indicate the -0.25 threshold for potentially damaging dynamic effects. Subplot b) provides a close-up of data falling to the left of this threshold.

load. In contrast to the behaviour seen in rolling direction results, transverse direction ratio values are considerably higher, with a significant number of points lying below -0.25 and above 0.25. This is attributable to the high level of conformity between rollers and raceways in this direction, meaning contact conjunction edges move rapidly outwards/inwards as the applied load

varies. With regards to film thickness and bearing damage it is negative ratio values which are of interest; this is because it is when load is rapidly removed (and so the contact patch shrinks) that local reductions in film thickness are known to occur, increasing the risk of damage. Figure 5a therefore includes a vertical dashed line indicating the location of the -0.25 threshold. A close-up of data falling to the left of this line is then shown in Figure 5b. Various features of these plots are worth considering, the most immediately evident being that maximum observed ratio values (in terms of magnitude) reduce

as roller load increases. This is expected from the Hertzian equations: since $a \propto w^{1/3}$ (where $w$ is roller load) it follows that $\dot{a} \propto \dot{w}/w^{2/3}$, a quantity for which the same value of $\dot{w}$ seen at a higher load elicits a smaller $\dot{a}$ response. The implication of this behaviour is that the possibility of dynamic effects is highest for lower values of roller load, with the largest ratio values seen close to the smallest loads (above the 1kN cutoff). However, $\dot{a}/\tilde{u}$ values less than -0.25 are seen with moderate levels of applied loading, including one for which $w = 12.5$kN. Since the contact conditions dataset represents only a subset of the

total simulation data, which itself contains a total of 30 10-minute duration simulations, higher load values in this region could well be possible. Presented results therefore indicate that, in the transverse direction, dynamic EHL effects are predicted to be taking place, but at loads in approximately the lowest 10% of those observed across the operational envelope. The impacts of such effects on the lubrication film and possible damage for this main-bearing are not yet known. Furthermore, dynamic effects are predicted to occur in the transverse, as opposed to rolling, direction. Very little, if any, work has been undertaken

which investigates EHL in these circumstances. While consideration has been given to side-leakage effects associated with ellipticity (Wheeler et al., 2016; Damiens, 2003), the authors are unaware of any work which considers transverse direction



(only) dynamic behaviour. This may be due to the relatively unique conditions experienced by wind turbine main-bearings, meaning such behaviour in rolling element bearings may not have been observed/predicted before now.

It is emphasised that there is uncertainty present in these results due to modelling approximations and assumptions, as well as the use of finite differencing to estimate gradients. In particular, possible sliding/skidding behaviour in a real main-bearing will influence mean entrainment velocity values, $\tilde{u}$, along with the time variations of load seen by each roller. Influences of main-bearing housing elasticity, not accounted for in this model, may also alter the magnitude and distribution of load around the bearing. Note, due to the $\dot{a} \propto \dot{w}/w^{2/3}$ relationship, inclusion of housing elasticity could lead to increases or decreases in dynamic behaviour, more work would be needed to determine its effect. Finally, further work is required to understand the interactions of these predicted dynamic effects within a grease lubricated (most probably starved) full roller bearing. In particular, it is necessary to consider what the effects of rapidly elongating contact patches (transverse to rolling) may be with regards to the dispersion/distribution of grease/bled-oil, both inside of individual contacts and throughout the main-bearing as a whole.

## 5    Discussion and Conclusions

There is much to unpack in the results which have been presented. A summary of key findings is therefore provided prior to further discussion:

1. The spherical roller main-bearing contacts in question should be treated as equivalent line contacts for the purposes of film thickness analysis using simplified lubrication equations

2. When treated as such, lubrication conditions (in terms of dimensionless parameters) fall well within the region over which the applied equation was fitted (except at the very lowest load levels)

3. Maximum pressures estimated to occur in the contact conjunctions were lower than might be expected, with values not exceeding about 1.6 GPa. More complex effects, not accounted for here, could lead to the occurrence of higher pressure values than has been estimated

4. In a best case scenario of fully-flooded lubrication, $T = 35\,°C$ was found to represent a transition point between EHL and mixed lubrication regimes. By $T = 40\,°C$, close to 80% of fully-flooded operating points fell into the mixed lubrication regime

5. Since main-bearings are grease lubricated, starved conditions are expected to be present, but, levels of starvation unknown. Starvation estimates taken from the literature (around a 30% reduction in film height) were applied, the effect of which was dramatic. Under the assumed level of starvation, close to 90% of operating points for $T = 35\,°C$ fell into the mixed regime, along with all operating points for $T = 40\,°C$. Importantly, more severe levels of starvation could be present in real wind turbine main-bearings



6. A sensitivity analysis revealed that results were most impacted by the temperature, starvation level and $\alpha^*$ value. The effect of roller load on film thickness results was small

7. Based on available information in the literature, it was tentatively concluded that significant levels of grease-thickener interactions (with the contact conjunction) appear unlikely for the modelled main-bearing

8. Dynamic EHL effects were predicted to be negligible in the direction of rolling, but potentially of significance in the transverse direction. Possible dynamic effects in this latter case were found to occur at lower loads, with observed maximum values around 12.5 kN

As has been emphasised, uncertainties are present which mean that care must be taken when interpreting these findings. In particular, further work is needed to better understand properties related to key sensitivities, these being temperature, starvation and $\alpha^*$ values, for wind turbine main-bearings and their lubricating greases. A potentially important consideration is also that, while presented results indicate lubricating films are only weakly sensitive to load, loading behaviour could still prove to have a significant but *indirect* effect. Specifically, it will be important to consider whether the highly variable and structured loading experienced by main-bearings (Hart, 2020) may influence grease dispersion/distribution within the bearing and/or frictional behaviour such that starvation levels and inlet temperatures are impacted. It will also be necessary to determine the impact of effects not accounted for here, including housing and bedplate flexibility and dynamic roller behaviour (skidding, skewing etc.). Dynamic EHL effects in the main-bearing also warrant further attention, especially when considering that the case of transverse only dynamic effects has not been studied in detail previously.

The modelled main-bearing under starved conditions was predicted to experience some proportion of time operating in the mixed lubrication regime for all temperatures of, $T \geq 30$ °C. If starvation is more severe in reality, mixed lubrication may begin at lower temperatures. These findings imply that the modelled main-bearing would be operating under increased levels of friction and a heightened risk of wear/micro-pitting for a non-negligible proportion of its operational life. This result has clear implications for the likelihood of damage associated with such conditions, but, is also relevant to fatigue life estimates undertaken during wind turbine design/component-selection stages. Rolling bearing fatigue life predictions are made under the assumption that fully EHL conditions hold throughout the bearing lifetime (Hart et al., 2020). It may therefore be the case that this assumption does not hold for wind turbine main-bearings. If so, it is not clear at this stage how this would effect the validity/accuracy of main-bearing fatigue life estimates.

With all described caveats in place, it is concluded that further development of the scientific understanding of lubrication in wind turbine main-bearings is a necessary part of ongoing efforts to identify and understand the key drivers of observed high rates of failure for this component.



## Appendix A: Table of symbols

| | |
|---|---|
| $a$ | semi-major contact dimension, for the modelled bearing this lies transverse ($y$) to the rolling direction (m) |
| $b$ | semi-minor contact dimension, for the modelled bearing this lies in ($x$) the rolling direction (m) |
| $E_I, E_{II}$ | Youngs modulii of solids I and II (Pa) |
| $E'$ | reduced modulus of elasticity (Pa), $2/E' = (1 - \nu_I^2)/E_I + (1 - \nu_{II}^2)/E_{II}$ |
| $G$ | dimensionless material parameter (-), $G = \alpha^* E'$ |
| $h, h_m, h_c$ | film thickness, minimum film thickness and central films thickness, respectively (m) |
| $k$ | ellipticity parameter (-), $k = a/b$ |
| $L$ | dimensionless viscosity parameter (Moes) (-), $L = G(2U)^{1/4}$ |
| $M_l, M$ | dimensionless load parameter (Moes) for line and point contacts respectively (-), $M_l = W_l(2U)^{-1/2}$, $M = W(2U)^{-3/4}$ |
| $p$ | pressure (Pa) |
| $r_{Ix}, r_{IIy}, \dots$ | radius of curvature of surface I/II in the $x/y$ direction, a strictly positive quantity (m) |
| $R_x$ | reduced radius of curvature in the entrainment direction (m), $\frac{1}{R_x} = \frac{sgn(Ix)}{r_{Ix}} + \frac{sgn(IIx)}{r_{IIx}}$ |
| $R_y$ | reduced radius of curvature transverse to the entrainment direction (m), $\frac{1}{R_y} = \frac{sgn(Iy)}{r_{Iy}} + \frac{sgn(IIy)}{r_{IIy}}$ |
| $sgn(\cdot)$ | $sgn(Ix)$ is 1 if surface I is convex in the $x$-direction and $-1$ if it is concave in the $x$-direction (similarly for II and/or $y$) |
| $s_l, s_p$ | surface stress in line and point contacts, respectively (Pa) |
| $s_{std}$ | standard deviation of surface heights (assuming normally distributed roughness, $s_{std} = \sigma$) (m) |
| $T$ | lubricant inlet temperature (°C) |
| $u_I, u_{II}$ | tangential velocities, in the entrainment direction ($x$), of surfaces I and II at the contact location (m/s) |
| $\tilde{u}$ | mean entrainment velocity (m/s), $\tilde{u} = (u_I + u_{II})/2$ |
| $U$ | dimensionless speed parameter (-), $U = \mu_0 \tilde{u}/(E' R_x)$ |
| $v_I, v_{II}, \tilde{v}$ | similar to "$u$" terms, but transverse ($y$) to entrainment direction (m/s) |
| $V$ | dimensionless hardness number, $V = v_h/E'$, for $v_h$ the surface Vickers hardness (-) |
| $w$ | normal load in point contact (N) |
| $w_l$ | normal load per unit length in line contact (N/m) |
| $W_l, W$ | dimensionless load parameter for line and point contacts respectively (-), $W_l = w_l/(E' R_x)$, $W = w/(E' R_x^2)$ |
| $\alpha$ | pressure-viscosity coefficient of the lubricant (at the inlet temperature, $T$) (Pa$^{-1}$) |
| $\alpha^*$ | inverse asymptotic isoviscous pressure coefficient (at the inlet temperature, $T$) (Pa$^{-1}$), $1/\alpha^* = \int_0^\infty \frac{\mu_0}{\mu} dp$ |
| $\Lambda$ | lubrication film parameter (-), $\Lambda = h_m/\sigma$ |
| $\mu$ | lubricant dynamic viscosity (Pa$\cdot$s) |
| $\mu_0$ | lubricant dynamic viscosity at the inlet temperature, $T$, and for (gauge-pressure) $p = 0$ (Pa$\cdot$s) |
| $\nu_I, \nu_{II}$ | Poisson's ratios of solids I and II (-) |
| $\rho$ | lubricant density (kg$\cdot$m$^{-3}$) |
| $\rho_0$ | lubricant density at the inlet temperature, $T$, and for (gauge-pressure) $p = 0$ (kg$\cdot$m$^{-3}$) |
| $\sigma_I, \sigma_{II}$ | surface roughness, in the form of root-mean-square deviations, for surfaces I and II respectively (m) |
| $\sigma$ | combined roughness of contacting surfaces (m), $\sigma = \sqrt{\sigma_I^2 + \sigma_{II}^2}$ |
| $\Phi_l, \Phi_p$ | film thickness modification factors accounting for surface roughness effects, line and point contacts respectively (-) |



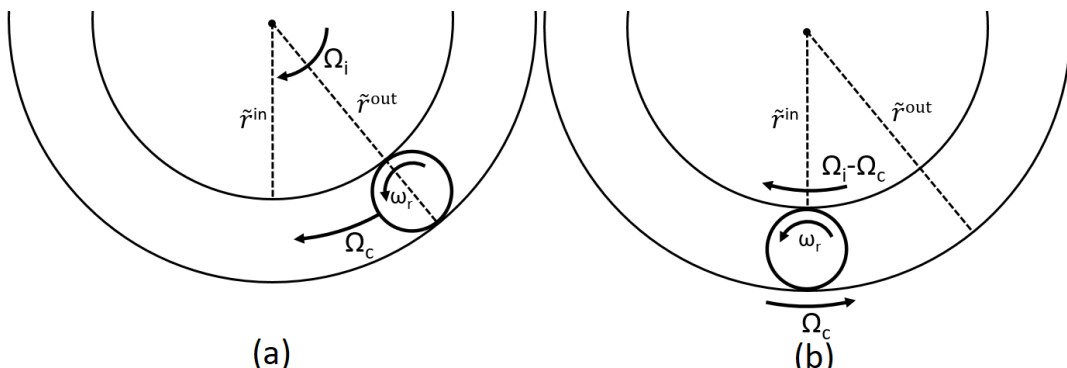

**Figure B1.** Roller orbiting bearing centre with respect to (a) stationary and (b) rotating reference frames; in the latter case the position of the roller centre remains fixed. $\Omega_i$ and $\Omega_c$ describe angular velocity with respect to the bearing centre, $\omega_r$ describes angular velocity with respect to the roller centre.

## Appendix B: Entrainment velocity under pure rolling

The mean entrainment velocity, $\tilde{u}$, is given by (Hart et al., 2021),

$$\tilde{u} = \frac{u_{\mathrm{I}} + u_{\mathrm{II}}}{2}, \tag{B1}$$

where $u_{\mathrm{I}}$ and $u_{\mathrm{I}}$ are surface velocities of bodies I and II (roller and raceway, respectively) in the direction of rolling. These velocities are dependent on shaft speed and system geometry. Figure B1 shows the cross-section of a roller orbiting the bearing centre in fixed and rotating reference frames, the latter moving with the roller centre. $\tilde{r}^{\mathrm{in}}$ and $\tilde{r}^{\mathrm{out}}$ are the perpendicular distances between the axis of rotation and inner- and outer- raceway contact centres (see Hart (2020)). Expressions for surface velocities may be derived as follows: Assuming pure rolling, inner and outer contact surface tangential velocities must be equal ($u_{\mathrm{I}} = u_{\mathrm{II}}$), hence,

$$\omega_r \left( \frac{\tilde{r}^{\mathrm{out}} - \tilde{r}^{\mathrm{in}}}{2} \right) = (\Omega_i - \Omega_c)\, \tilde{r}^{\mathrm{in}} \tag{B2}$$

$$\omega_r \left( \frac{\tilde{r}^{\mathrm{out}} - \tilde{r}^{\mathrm{in}}}{2} \right) = \Omega_c \tilde{r}^{\mathrm{out}}, \tag{B3}$$

for $\omega_r$, $\Omega_i$ and $\Omega_c$ the angular velocities shown in Figure B1. From this it follows,

$$\Omega_c = \Omega_i \left( \frac{\tilde{r}^{\mathrm{in}}}{\tilde{r}^{\mathrm{in}} + \tilde{r}^{\mathrm{out}}} \right). \tag{B4}$$

Since $u_{\mathrm{I}} = u_{\mathrm{II}}$,

$$\tilde{u}^{\mathrm{in}} = (\Omega_i - \Omega_c)\, \tilde{r}^{\mathrm{in}}$$
$$= \Omega_i \left( \frac{\tilde{r}^{\mathrm{in}} \tilde{r}^{\mathrm{out}}}{\tilde{r}^{\mathrm{in}} + \tilde{r}^{\mathrm{out}}} \right), \tag{B5}$$

$$\tilde{u}^{\mathrm{out}} = -\Omega_c \tilde{r}^{\mathrm{out}}$$
$$= -\Omega_i \left( \frac{\tilde{r}^{\mathrm{in}} \tilde{r}^{\mathrm{out}}}{\tilde{r}^{\mathrm{in}} + \tilde{r}^{\mathrm{out}}} \right). \tag{B6}$$



Entrainment velocities for inner and outer contacts can be seen to have equal magnitude but opposite direction in the applied frame of reference. Since lubrication is considered locally at each contact location, directional differences in the global reference frame are not relevant. Entrainment velocities at inner- and outer-raceways may therefore be considered equal, with both given by,

$$\tilde{u} = \Omega_i \left( \frac{\tilde{r}^{\mathrm{in}} \tilde{r}^{\mathrm{out}}}{\tilde{r}^{\mathrm{in}} + \tilde{r}^{\mathrm{out}}} \right). \tag{B7}$$

**Appendix C: The validity of modelling approximations**

Outputs were checked to provide confidence in the validity of approximations applied at modelling stages. For a more detailed discussion of the approximations themselves, see Hart et al. (2021). With regards to contact dimensions and curvature radii, values of $b/R_x$ at inner and outer contacts never exceed 3% and values of $a/R_y$ never exceed 0.3%. Figure C1 shows values of $h_m/\delta$ and $\delta/b$, where $\delta$ is roller deflection as given by Hertzian equations (see Hart (2020)), seen across the contact conditions dataset. These results show that maximum $h_m/\delta$ values are around 6%, but only at very low loads. At higher loads these values fall rapidly to around 0.2% and lower. From the $\delta/b$ results it follows that everywhere $\delta < b < a$, with the $h_m/\delta$ results therefore providing an upper limits for ratios involving these other contact dimensions. $h_m/\delta$ results demonstrate directly the validity of evaluating internal deflections and loading independently of lubrication, especially considering the results seen here for more highly loaded rollers, since, it will be these rollers which have most influence on the resulting internal loads and displacements at each timestep. From analysis of the presented results it may also be concluded that the "half-space" and "lubrication" approximations may be considered valid here.

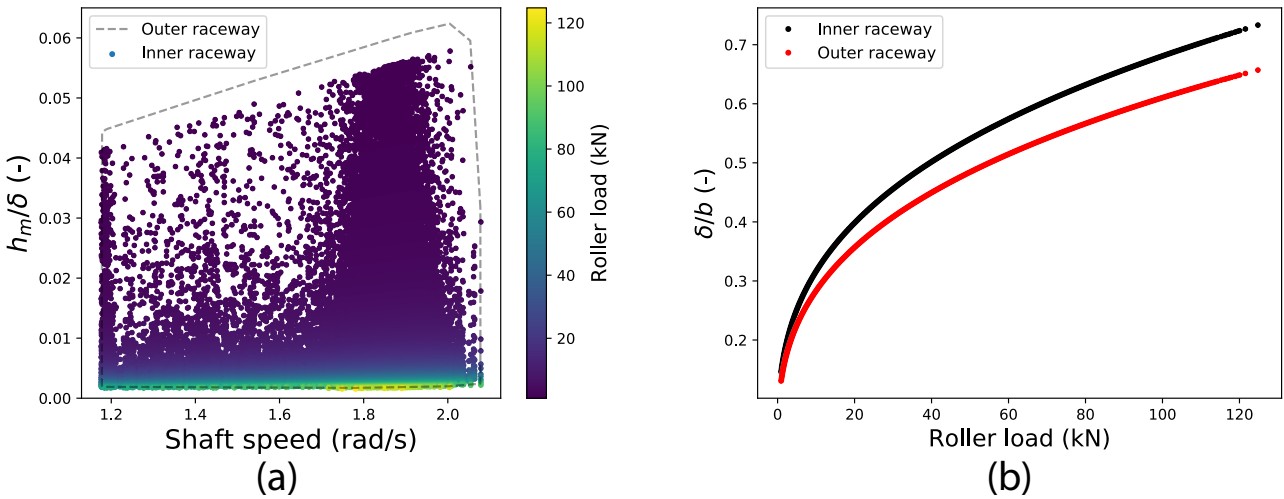

**Figure C1.** Contact dimension ratios plotted against shaft speed and/or roller load values.



## Appendix D: Lubrication results sensitivities

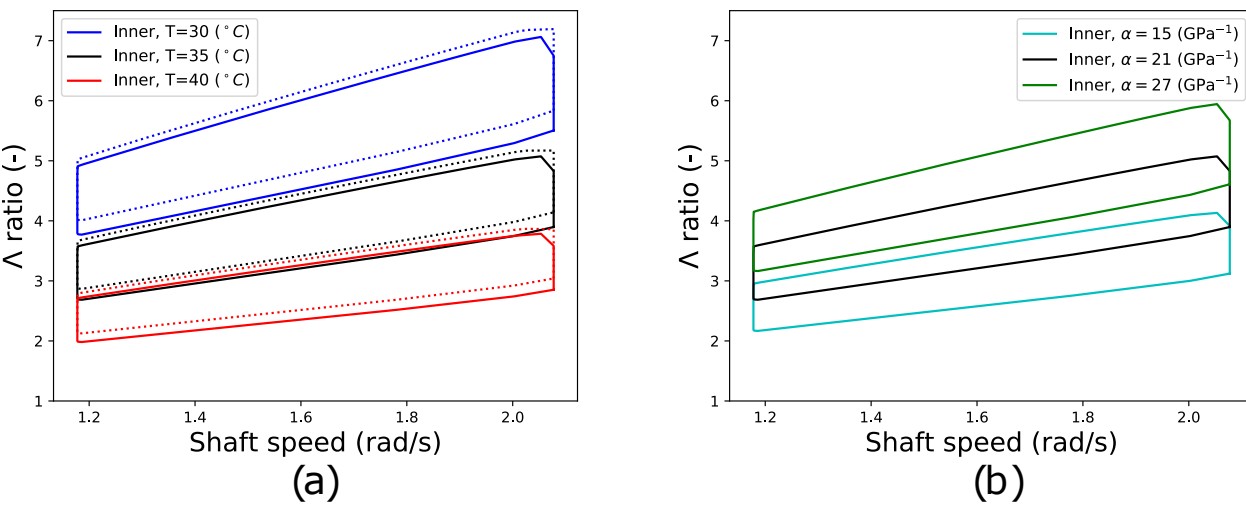

**Figure D1.** Graphical results for sensitivity analyses on load and $\alpha^*$ values. Subplot a) shows original results as solid lines, along with results obtained after all roller load values are halved (dotted lines). All three temperatures levels are included. Subplot b) shows the "standard case", along with results obtained when $\alpha^*$ is increased and decreased by $6\,\mathrm{GPa}^{-1}$.

*Competing interests.* The authors declare they have no competing interests.





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
