# Peer review of "Wind turbine main-bearing lubrication - Part 2: Simulation based results for a double-row spherical roller main-bearing in a 1.5 MW wind turbine"

_Wind Energy Science, 2021_

## Referee Comment (RC1)

**Review: Wind turbine main-bearing lubrication - Part 2: Simulation based results for a double-row spherical roller main-bearing in a 1.5 MW wind turbine**

Hart et al.

The manuscript illustrates the application of EHD concepts explained in Part 1 to a practical case of a spherical roller bearing in a main shaft of a wind turbine. Properties of a commercial grease for this application are used as input parameters and the calculation includes starvation effects variable contact load plus potential dynamic effects. The manuscript shows that the contacts experience mixed-lubrication regime in a large proportion of the running time.

The reviewer is favourable to the publication of the manuscript but requires some minor revisions or at least the answer of some questions.

Revisions:

1. Section 3.1: It is very interesting to see the source of the load and speed cases analysed in this study. The authors refer to the IEC design standards. Can the authors point out exactly which standard they use (number?).

2. Section 3.2: Perhaps the only missing aspect is to say that the considered properties of the grease are taken when the grease is fresh. It is well known that these properties change as the grease ages, but also the bleeding rate, so starvation will depend on the age of the grease somehow. The reviewer assumes that the authors did not consider re-greasing intervals in their model.

3. Section 3.2.1: What about cold starts or cold weather in Wind Turbines?

4. Section 4.2: For an EHL person $\Lambda$ values are a good way to sense the mean lubrication conditions in a contact. However, for bearings the ISO standards use $\kappa$ (defined in ISO 281). In the same standard an approximation between the two parameters is given $\kappa \approx \Lambda^{1.3}$, therefore it is possible to give order of magnitude of the results also in terms of $\kappa$. Notice that if this parameter is known, bearing life estimations are possible (also using ISO 281) and at least relative life values of different lubrication conditions can be obtained.

5. Section 4.2: Indeed the consideration of starvation in the present manuscript is somehow a bit disappointing. However, the reviewer understands that the modelling of starvation in greased lubricated bearings is not simple. Especially when the availability of grease in the contacts is unknown and also the aging status of the grease (bleeding rate). Besides all these in this application the bearings should (in general) be fully packed of grease, which means that there is also a "gravity" effect. The grease in the lowest part of the bearing might have to carry the weight of all other grease making the bleeding non-uniform.

6. Section 4.3: Grease thickener interactions, perhaps a more accurate calculation could be done with models: Nogi, (https://doi.org/10.1080/10402004.2020.1778147) and Morales-

Espejel (Tribology International 74 (2014) 7–19). Indeed, a worked grease will see reduced these benefits, but how much?

Section 4.3: Perhaps a more significant effects are stand-still periods and accelerations + decelerations. Sudden changes of speed. Stormy weather.

Section 5: The authors have written the following text "Rolling bearing fatigue life predictions are made under the assumption that fully EHL conditions hold throughout the bearing lifetime". In the experience of this reviewer this is inaccurate. Actually the parameter $\kappa$ takes into account the lubrication conditions as part of the factor $a_{ISO}$ in the standard ISO 281. Modern bearing life models can model variable operating conditions within the life of a bearing, not only speed but also load. Besides this, new bearing life models are being developed to explicitly separate surface and subsurface failure modes, the authors can look for such references (the reviewer has seen this at least in some SKF publications).

---

## Referee Comment (RC2)

**Reviewer Blind Comments to Author**

The submitted paper discuss the lubricating conditions of main-bearings of wind turbines using a double-row spherical roller bearing in a 1.5 MW wind turbine as case study. In the reviewer's opinion the submitted paper is well written and structured. However, the scientific significance of the applied approach is limited. Understandably the authors apply known methods and a significant number of assumptions, however, this leads to the question of how the proposed approach differs from the typical industrial approach.

The authors focus on reaching a wider readership by using simplified descriptions and methods. While this approach seems reasonable for part 1, it leads to the major short-coming of part 2: apparent missing novelty. In the reviewer's opinion the submitted paper could be shortened significantly and complemented with further aspects such as: consideration of sliding (due to the formation of a load zone, due to low loads and along the contact line - Heathcote slip), in-depth analysis of temperature distribution, lubrication type, geometry (such as modern tapered main-bearings), starvation….

In the reviewer's opinion the submitted paper is a case study rather than a research paper.

Please consider following points:

**Abstract**

- The temperature should be introduced as an influencing factor.

**Introduction and Background**

- The authors state that
  - *"higher than expected failure rates"* and *"premature failures in main-bearings"* occur. Please quantify and give the necessary references.
  - *"it cannot currently be known a priori whether main-bearing failure rates are likely to be improved or worse"*. In the reviewer's opinion the authors could discuss whether the contact conditions would most likely positively or negatively impact the lubricating conditions (speed, load, time between contacts and temperature could be compared).
  - *"fatigue life assessment……explicitly assumes EHL"*. Modern lifetime calculations do not assume EHL. For example, the $a_{iso}$ or the $a_{skf}$ consider the lubricating conditions through the viscosity ratio Kappa.
  - *"the described load structures were found to drive large, rapid variations.."*. Please quantify.
- Based on the investigations in Guo et al. (2021) the authors state that axial motions are slow and highly unlikely to impact the lubricating film. However, they state that *"the lubrication was not modelled directly"*. In this case, how was the aforementioned conclusion drawn?
- Sentences 29 through 38 are a good example of strongly simplified explanations which could be strongly shortened
- Please give the coordinates of the two radial components of the load vector

**Methodology**

- Please use references where is appropriate (e.g. IEC design standards)
- Could the authors please state if *"10 min."* refer to simulation or operation time?
- The authors explain that the applied force was considered (instead of the reaction force). Furthermore, they explain that it is necessary to ensure the correct one is calculated with the respect to the reference frame. How do the results will deviate if the reaction force is used instead? They have equal magnitude
- Please comment why the upwind row is *"only occasionally loaded during normal operations"*. One would expect that the radial load is carried by both rows
- In the reviewer's opinion the authors should expand their consideration of roller sliding. It is well known that sliding have a great impact on the contact temperature and the film formation
- The authors comment that *"other details are proprietary"*. In the reviewer's opinion profile and roughness measurements can be carried out and consider in half-space calculations
- Table 1
  - The combine surface-roughness seems high. Please give the individual values and the method used for determining the roughness
  - Please comment on the profile in width-direction
  - *"Inlet temperature"* is commonly used for "lubrication nozzles". Please consider writing "contact inlet"
- In the reviewer's opinion the selected temperatures could be too low. If the outer bearing casing is at 20-40 °C degrees the contact temperature (most likely) won't be in the range between 30 - 40°C. How did the authors make this assumption?
- Sentences 177 through 185 are a further example of strongly simplified explanations which could be strongly shortened. In which cases would the authors consider a spherical-roller bearing as a point contact (aside from insignificant loads)?
- One could argue that the consideration of starvation is not scientifically accurate. In the reviewer's opinion the authors should either select a concrete approach or take the sub-chapter out. If the authors decide to expand this chapter please consider modern literature regarding starvation and the influence of base oil and thickener on the film formation.

**Results**

- Sentences 231 through 240 are a further example of strongly simplified explanations which could be strongly shortened.
- Please consider using the shaft speed instead of the wind speed for Figure 10.
- In the discussion of Figure 10 the authors state that around the bearing circumference an unloaded region is generally included. While this is true for radial loads the authors stated before, that the down-wind bearing was used as case study. This bearing is supposed to be axially loaded, which could result in a load zone over 360°.
- The authors explain that *"outlying points at very low loads correspond to high values of film thickness"*. Please mark this points int Figure 10

- Please revise sentence 254-255. It gives the impression that the load is the determining influencing factor
- It is commonly known that the pressure increases in the area of the PETRUSEVICH-peak. How do the authors assume that the maximum EHL pressure values are equivalent to those seen in dry contacts?
- Did the authors consider the axial load in the pressure calculations?
- Please comment whether the minimal loading of the bearing is guaranteed and how this affect its sliding behavior
- Please give the used formula for lambda
- Please comment if the selected variations step in table 2 are plausible. Furthermore, the magnitude of pressure influence is dependent on the load magnitude (as shown in Fig 3.), therefore, the simplified statement that by reducing the load by 50 % the lambda value increases by 6 % is misleading.
- Why did the authors consider the bearing as a point contact for chapter 4.3?
- Why was a threshold of 25 % selected? Please add references

**Discussion and conclusions**
- The authors should discuss the influence of running-in
- No further comments. Please consider the points above

---

## Author Comment (AC1)

Wind turbine main-bearing lubrication - Part 2: Simulation based results for a double-row roller main-bearing in a 1.5 MW wind turbine

**Response to reviewer 1**

Dear reviewer,

First, we would like to thank you for assessing our manuscript and suggesting improvements. We will make ensure the updated manuscript includes the suggestions you have made here.

A detailed response is now provided. We include your comments below in blue, followed by our responses in **black**.

The manuscript illustrates the application of EHD concepts explained in Part 1 to a practical case of a spherical roller bearing in a main shaft of a wind turbine. Properties of a commercial grease for this application are used as input parameters and the calculation includes starvation effects variable contact load plus potential dynamic effects. The manuscript shows that the contacts experience mixed-lubrication regime in a large proportion of the running time.

The reviewer is favourable to the publication of the manuscript but requires some minor revisions or at least the answer of some questions.

Revisions:

1. Section 3.1: It is very interesting to see the source of the load and speed cases analysed in this study. The authors refer to the IEC design standards. Can the authors point out exactly which standard they use (number?).

Yes indeed, we are referring to IEC 61400-1:2019 "Wind energy generation systems - Part 1: Design requirements". We also agree that this should be made clear in the manuscript itself and so will reference this standard explicitly in the revised version of the paper.

2. Section 3.2: Perhaps the only missing aspect is to say that the considered properties of the grease are taken when the grease is fresh. It is well known that these properties change as the grease ages, but also the bleeding rate, so starvation will depend on the age of the grease somehow. The reviewer assumes that the authors did not consider re-greasing intervals in their model.

This is a good point which we will add into the revised manuscript. Yes, you are correct that for this analysis we did not consider re-greasing intervals, but that is something we are hoping to do in future work. Indeed we will indicate this is the discussion at the end of the paper to help make important future work in this area clear.

3. Section 3.2.1: What about cold starts or cold weather in Wind Turbines?

Excellent point, we will include some discussion of this when revising the manuscript and we'll refer to papers which are being added to the "Part 1" review paper on start-stop effects as well as lubricant redistribution when stationary.

4. Section 4.2: For an EHL person $\Lambda$ values are a good way to sense the mean lubrication conditions in a contact. However, for bearings the ISO standards use $\kappa$ (defined in ISO 281). In the same standard an approximation between the two parameters is given $\kappa \approx \Lambda 1.3$, therefore it is possible to give order of magnitude of the results also in terms of $\kappa$. Notice that if this parameter is known,

bearing life estimations are possible (also using ISO 281) and at least relative life values of different lubrication conditions can be obtained.

This is an excellent point and we agree that the link to $\kappa$ (defined in ISO 281) is relevant to include in the paper to give the reader a clear picture for how these things link together. We will therefore include this discussion in the revised manuscript.

5. Section 4.2: Indeed the consideration of starvation in the present manuscript is somehow a bit disappointing. However, the reviewer understands that the modelling of starvation in greased lubricated bearings is not simple. Especially when the availability of grease in the contacts is unknown and also the aging status of the grease (bleeding rate). Besides all these in this application the bearings should (in general) be fully packed of grease, which means that there is also a "gravity" effect. The grease in the lowest part of the bearing might have to carry the weight of all other grease making the bleeding non-uniform.

Yes it would have been nice to provide a more detailed assessment of starvation. Unfortunately, this is not possible at the current stage, even when considering more recent work on starvation which has now been added to the "Part 1" review. We are hoping that this work serves as motivation for more detailed analyses with the kinds of sophisticated models that could try and capture such things, or alternatively laboratory experiments – some of which we are planning on performing ourselves.

6. Section 4.3: Grease thickener interactions, perhaps a more accurate calculation could be done with models: Nogi, (https://doi.org/10.1080/10402004.2020.1778147) and Morales-Espejel (Tribology International 74 (2014) 7–19). Indeed, a worked grease will see reduced these benefits, but how much?

Agreed, and we're hoping that we might manage such modelling in the future. But, unfortunately it is not something we were able to undertake in the current work.

Section 4.3: Perhaps a more significant effects are stand-still periods and accelerations + decelerations. Sudden changes of speed. Stormy weather.

Yes these types of events could well be important here. We will include more discussion of them in the revised manuscript and suggestions for future work building on what we have done here.

Section 5: The authors have written the following text "Rolling bearing fatigue life predictions are made under the assumption that fully EHL conditions hold throughout the bearing lifetime". In the experience of this reviewer this is inaccurate. Actually the parameter $\kappa$ takes into account the lubrication conditions as part of the factor $aISO$ in the standard ISO 281. Modern bearing life models can model variable operating conditions within the life of a bearing, not only speed but also load. Besides this, new bearing life models are being developed to explicitly separate surface and subsurface failure modes, the authors can look for such references (the reviewer has seen this at least in some SKF publications).

Thank you for pointing out this oversight on our part. We will make sure to correct this in the revised paper and give a more accurate overview of the life assessment of ISO 281.

---

## Author Comment (AC2)

Wind turbine main-bearing lubrication - Part 2: Simulation based results for a double-row spherical roller main-bearing in a 1.5 MW wind turbine

**Response to reviewer 2**

Dear reviewer,

First, we would like to thank you for your considerable efforts in assessing our manuscript and suggesting improvements. We will make ensure the updated manuscript includes the suggestions you have made here.

A detailed response is now provided. We include your comments below in **blue**, followed by our responses in **black**.

The submitted paper discuss the lubricating conditions of main-bearings of wind turbines using a double-row spherical roller bearing in a 1.5 MW wind turbine as case study. In the reviewer's opinion the submitted paper is well written and structured. However, the scientific significance of the applied approach is limited. Understandably the authors apply known methods and a significant number of assumptions, however, this leads to the question of how the proposed approach differs from the typical industrial approach.

In response to this point, and some others below, it is important to discuss the aims and position of this paper in the context of the literature for main-bearings in wind turbines. The main-bearing is a component which has been identified as a problem for the wind industry, but one for which the literature has been relatively sparse until recently. A growing number of studies have now considered various aspects of the operation and reliability of this components, but, before now there have been no papers whatsoever which consider lubrication and EHL conditions for this component. Since the main-bearing is a tribological and lubricated component this aspect of its operation must be included if the premature failures seen in the field are to be properly understood and ultimately prevented. The difficulty when it comes to understanding this component is the fundamentally multi-disciplinary nature of the problem, since it is a bearing affixed to a flexible structure whose input loads are complex and determined by interactions between the wind turbine rotor and turbulent wind fields. Bringing these various aspects of the problem together necessitates simpler analyses to begin with, in order for important aspects and interactions to be identified – allowing areas for further analysis (using more sophisticated models) to be identified and prioritised. This paper represents that first effort for main-bearings in wind turbines. As such, we don't necessarily claim that we are doing things very much differently from the current industrial approach, but, the methods we use are applied transparently and with careful consideration of their validity and limitations etc. Furthermore, the industrial work on main-bearings you refer to isn't available in the public domain and so can't be built on as ours can. Finally, as with the review portion of the paper, a principal aim here is also the promotion of interdisciplinary understanding for engineers in related disciplines to be able to perform similar analyses and understand how our results impacts them. The current paper also achieves this goal and is the first to do this (for the main-bearing), this is another aspect of the paper's novelty, in addition to the new knowledge being presented here for this component. Overall we believe that this paper presents a solid foundational set of results, using existing methods applied with careful consideration, which lays the groundwork for further work and more sophisticated analyses.

The authors focus on reaching a wider readership by using simplified descriptions and methods. While this approach seems reasonable for part 1, it leads to the major short-coming of part 2: apparent missing novelty. In the reviewer's opinion the submitted paper could be

shortened significantly and complemented with further aspects such as: consideration of sliding (due to the formation of a load zone, due to low loads and along the contact line - Heathcote slip), in-depth analysis of temperature distribution, lubrication type, geometry (such as modern tapered main-bearings), starvation….

As described above we believe these is important novelty in this paper already. In future work we are going to be working on some of the things you mention here, but they fall outside the aims and context of this work. For example, wind turbines never see steady operating conditions, as such analysis of slip requires a fully dynamic main-bearing model. Such a model will require significant development beyond our current capabilities and so falls out of scope for this work at present. Similarly, an in-depth analysis of the temperature distribution would require thermal modelling of the main-bearing is not yet available in our model. With respect to temperature we know that the temperature range we study is close to the values seen in practise and, importantly, the results indicate that these temperatures may be seen as a change point between lubrication regimes. This result is valuable and important in of itself and indicates where future analysis should be focussed. As stated above, we believe the results presented provide an important baseline on which to build. We agree that the analyses you suggest should be undertaken, but we believe they fall into the category of 'recommended future work'.

In the reviewer's opinion the submitted paper is a case study rather than a research paper.
Since this paper focusses on an individual wind turbine and a specific main-bearing it is perhaps fair enough to consider it a 'case study'. I don't agree that this means it is not also a 'research article'. As outlined above, this is the only article in the literature which has presented results of this nature for a wind turbine main-bearing, meaning it provides important research novelty despite focussing on one specific turbine and bearing.

Please consider following points:
Abstract
• The temperature should be introduced as an influencing factor.
We will add this as suggested.
Introduction and Background
• The authors state that
      o "higher than expected failure rates" and "premature failures in main-bearings" occur. Please quantify and give the necessary references.
We will add numbers and references to this when revising the paper
      o "it cannot currently be known a priori whether main-bearing failure rates are likely to be improved or worse". In the reviewer's opinion the authors could discuss whether the contact conditions would most likely positively or negatively impact the lubricating conditions (speed, load, time between contacts and temperature could be compared).
We will consider if more context/discussion can be given here
      o "fatigue life assessment……explicitly assumes EHL". Modern lifetime calculations do not assume EHL. For example, the aiso or the askf consider the lubricating conditions through the viscosity ratio Kappa.
Thank you for pointing out this error, we will correct when revising the paper
      o "the described load structures were found to drive large, rapid variations..". Please quantify.
We will add concrete numbers here
      • Based on the investigations in Guo et al. (2021) the authors state that axial motions are slow and highly unlikely to impact the lubricating film. However, they state that "the lubrication was not modelled directly". In this case, how was the aforementioned conclusion drawn?

In that work, the maximum axial speed of the rollers was expressed as a percentage of entrainment velocity, then this was compared with results in the literature regarding when the axial motion will disrupt the EHL film. Their % was lower than that given as the cutoff in the literature, hence they concluded that axial motions were unlikely to be impacting the lubricant film. We will make this clearer when revising the paper.

• Sentences 29 through 38 are a good example of strongly simplified explanations which could be strongly shortened

We will see about shortening this.

• Please give the coordinates of the two radial components of the load vector

Will do.

Methodology

• Please use references where is appropriate (e.g. IEC design standards)

We will add these in

• Could the authors please state if "10 min." refer to simulation or operation time?

Operational time, we will make this clear in the manuscript

• The authors explain that the applied force was considered (instead of the reaction force). Furthermore, they explain that it is necessary to ensure the correct one is calculated with the respect to the reference frame. How do the results will deviate if the reaction force is used instead? They have equal magnitude

This is important for tracking the loads on individual rollers as they move around the main-bearing. The Hertzian contact model takes an input of the applied load vector, with the vector direction determining the location of applied load etc. The rollers are simultaneously orbiting the bearing centre and so their positions relative to applied loading matters. If the reaction load is used the load will be of the correct magnitude, but acting in the wrong direction and so maximally loading the wrong rollers.

• Please comment why the upwind row is "only occasionally loaded during normal operations". One would expect that the radial load is carried by both rows

A wind turbine sees very large thrust loads, this causes axial deflections which tend to unload the upwind row throughout much of the operating time of the turbine. The second row is there mostly for when the turbine if not operating. This is shown in https://doi.org/10.1002/we.2549 and also provides motivation for the design changes proposed in https://doi.org/10.1007/s10010-021-00462-1

• In the reviewer's opinion the authors should expand their consideration of roller sliding. It is well known that sliding have a great impact on the contact temperature and the film formation

As mentioned above, a wind turbine never sees steady-state operating conditions and so standard sliding models cannot be used to evaluate this. A full dynamic model would be needed instead. This is something we intend to develop but not a capability we have at this time. Furthermore, it is our understanding that sliding will indeed affect the bearing temperature through friction, but that its effect on film thickness is relatively small because it is the viscosity (and hence temperature) of the lubricant at the contact inlet that most directly determines film height. So, the main effect of sliding would be to elevate the overall temperature of the main-bearing. But, this is something we already account for indirectly because the temperature range we use is based on measured temperatures of operating main-bearing casings (we'll revisit this topic in a later comment).

• The authors comment that "other details are proprietary". In the reviewer's opinion profile and roughness measurements can be carried out and consider in half-space calculations

We will consider if more information can be given.

• Table 1

   o The combine surface-roughness seems high. Please give the individual values and the method used for determining the roughness

We will need to check to see if individual values can be given. These are combined by taking the square-root of the sum of squares of the values.

o Please comment on the profile in width-direction

We will see if this is information we have available to us.

o "Inlet temperature" is commonly used for "lubrication nozzles". Please consider writing "contact inlet"

We will do this as suggested

• In the reviewer's opinion the selected temperatures could be too low. If the outer bearing casing is at 20-40 °C degrees the contact temperature (most likely) won't be in the range between 30 - 40°C. How did the authors make this assumption?

These are low speed bearings and so, based on information we could find on casing versus internal temperatures, we don't expect the internal temp to be much higher than that of the casing. In addition, and as discussed in the paper, the values are chosen because they are certainly within the normal operating range for a wind turbine main-bearing (based on available data) but also because they represent a transition point between lubrication regimes. Higher temperatures are possible, but this would only strengthen the conclusion of the paper that mixed lubrication is expected.

• Sentences 177 through 185 are a further example of strongly simplified explanations which could be strongly shortened. In which cases would the authors consider a spherical-roller bearing as a point contact (aside from insignificant loads)?

Our definition of point contact, as given in "Part 1", includes both circular and elliptical contacts. Therefore, yes we consider and SRB roller to be a point contact in the context of Hertzian theory. However, as shown in the paper, we argue that it should be treated as a line contact for the purposes of EHL film analysis.

• One could argue that the consideration of starvation is not scientifically accurate. In the reviewer's opinion the authors should either select a concrete approach or take the sub-chapter out. If the authors decide to expand this chapter please consider modern literature regarding starvation and the influence of base oil and thickener on the film formation.

We agree that this analysis of starvation is limited. But on reviewing the most recent available literature on starvation (as suggested by yourself in "Part 1" – where the starvation section is now much expanded), it remains the case that current analytical formulas are not yet at the stage where they may be practically applied to a full bearing. However, we believe that the analysis as presented still provides useful information by considering the impacts of film reductions of the general order of magnitude seen during starvation. This helps show that starvation is a key consideration for this main-bearing since it could be the different between full EHL and mixed lubrication. But on reflection we think that this part of the analysis could be better presented in the context we have described it here. Therefore, we will revise the description, motivations and interpretations of this part of the analysis carefully.

Results

• Sentences 231 through 240 are a further example of strongly simplified explanations which could be strongly shortened.

We will consider shortening this.

• Please consider using the shaft speed instead of the wind speed for Figure 10.

I assume you are referring to Fig 2. Windspeed is used as this is the variable that dictates the operating point of a wind turbine. The rotational speed of the wind turbine only changes for part of it's operating range and so would result in a less clear figure. Furthermore, the 'speed' from a lubrication point of view is represented by the dimensionless viscosity/speed parameter L. For these reasons we believe wind speed is the better candidate for the colorscale.

• In the discussion of Figure 10 the authors state that around the bearing circumference an unloaded region is generally included. While this is true for radial loads the authors stated before, that the down-wind bearing was used as case study. This bearing is

supposed to be axially loaded, which could result in a load zone over 360°.
Yes this is indeed correct, and 360 degree loading can happen, but it is more common that not that there will be an unloaded region on the bearing. This comment is there to explain why a full range of M (dimensionless load) values is seen across all wind speeds – and the abundance of M values of the left shows that unloaded rollers are common.

• The authors explain that "outlying points at very low loads correspond to high values of film thickness". Please mark this points int Figure 10
I am not sure if this needs to be pointed out on the figure itself, since the load is captured by M directly. But, we will make sure when revising the paper to specify those values more clearly as related to M. We will also consider whether to add anything to the figure for clarity.

• Please revise sentence 254-255. It gives the impression that the load is the determining influencing factor
Will do.

• It is commonly known that the pressure increases in the area of the PETRUSEVICH-peak. How do the authors assume that the maximum EHL pressure values are equivalent to those seen in dry contacts?
We use the Hertzian contact pressure as an estimate of maximum pressure in the conjunction, which is the case for heavily loaded rollers (we discuss this in "Part 1" of the paper). We will make this clearer in the manuscript.

• Did the authors consider the axial load in the pressure calculations?
Yes, pressure values were calculated after the Hertzian contact model was used to evaluate all rollers loads under both radial and axial loads applied to the bearing.

• Please comment whether the minimal loading of the bearing is guaranteed and how this affect its sliding behaviour
For an SRB in a wind turbine, minimal loading is not guaranteed as there is no preloading. As has been shown in previous work https://doi.org/10.1002/we.2549 unloading of the main-bearing can occur during operation. This would likely introduce sliding into the system but we are not yet able to quantify this effect.

• Please give the used formula for lambda
We will add this in.

• Please comment if the selected variations step in table 2 are plausible. Furthermore, the magnitude of pressure influence is dependent on the load magnitude (as shown in Fig 3.), therefore, the simplified statement that by reducing the load by 50 % the lambda value increases by 6 % is misleading.
Yes these are plausible changes in variable values based on the data available to us. We will look to make sure that the reasons for this are given in the revised manuscript. With respect to the pressure/load comment I believe you are referring to the fact that the change in film thickness with change in load is different at different loads. This is certainly true, although the same is also true for the other varied parameters, e.g. temperature, when they are changed. But, please note that the values given in the table are average values of changes in Lambda from sensitivity testing across the whole dataset of operating points. Therefore, this includes the fact that changes are different at around different operating points.

• Why did the authors consider the bearing as a point contact for chapter 4.3?
Again we should clarify that our definition of point contact includes both elliptical and circular contacts. Here we are treating it as a long elliptical contact. For lubrication it was treated as a line contact for reasons that were outlined. But in 4.3 we need to understand changes in its dimensions as an elliptical contact (changes in both a and b) since this is what it is. The reasons that we needed to use equivalent line contact equations before do not apply here, and so it was deemed appropriate and necessary to perform an elliptical contact analysis.

• Why was a threshold of 25 % selected? Please add references
We will add a reference for this.

Discussion and conclusions
• The authors should discuss the influence of running-in
We will look to include this in the revised paper.
• No further comments. Please consider the points above

---

## Author Comment (AC3)

Wind turbine main-bearing lubrication - Part 2: Simulation based results for a double-row spherical roller main-bearing in a 1.5 MW wind turbine

**Response to Editor**

Dear Amir,

Thank you for handling the review process for this paper.

We are grateful to the reviewers for providing excellent suggestions and feedback which will help improve this manuscript.

Reviewer 2 asked some legitimate questions concerning research novelty for Part 2 of this overall paper. We have provided a detailed discussion on this within our response to Reviewer 2, but felt it was helpful to include some of this here as well. The following is an excerpt from that response which reiterates the aims, context and novelty of this manuscript:

"The main-bearing is a component which has been identified as a problem for the wind industry, but one for which the literature has been relatively sparse until recently. A growing number of studies have now considered various aspects of the operation and reliability of this components, but, before now there have been no papers whatsoever which consider lubrication and EHL conditions for this component. Since the main-bearing is a tribological and lubricated component this aspect of its operation must be included if the premature failures seen in the field are to be properly understood and ultimately prevented. The difficulty when it comes to understanding this component is the fundamentally multi-disciplinary nature of the problem, since it is a bearing affixed to a flexible structure whose input loads are complex and determined by interactions between the wind turbine rotor and turbulent wind fields. Bringing these various aspects of the problem together necessitates simpler analyses to begin with, in order for important aspects and interactions to be identified – allowing areas for further analysis (using more sophisticated models) to be identified and prioritised. This paper represents that first effort for main-bearings in wind turbines. As such, we don't necessarily claim that we are doing things very much differently from the current industrial approach, but, the methods we use are applied transparently and with careful consideration of their validity and limitations etc. Furthermore, the industrial work on main-bearings you refer to isn't available in the public domain and so can't be built on as ours can. Finally, as with the review portion of the paper, a principal aim here is also the promotion of interdisciplinary understanding for engineers in related disciplines to be able to perform similar analyses and understand how our results impacts them. The current paper also achieves this goal and is the first to do this (for the main-bearing), this is another aspect of the paper's novelty, in addition to the new knowledge being presented here for this component. Overall we believe that this paper presents a solid foundational set of results, using existing methods applied with careful consideration, which lays the groundwork for further work and more sophisticated analyses."

The above outlines the important novelty of this manuscript for main-bearings in wind turbines and why we believe it is a valuable article for WES readers.

We look forward to submitting a revised version of the manuscript.

If anything in our responses requires further clarification, we'll be happy to provide it.

Edward Hart
(corresponding author)

---

## Author Response (AR1)

Wind turbine main-bearing lubrication - Part 2: Simulation based results for a double-row spherical roller main-bearing in a 1.5 MW wind turbine

**Response to Editor (updated post-revisions)**

Dear Amir,

Thank you for handling the review process for this paper.

We are grateful to the reviewers for providing excellent suggestions and feedback. We have now revised the manuscript in line with their suggestions. In most cases we have revised the paper exactly as they have requested. In any instances where we have done differently, a detailed discussion of our logic and thought process is provided. Therefore, in addition to this response to yourself, detailed and updated responses to both reviewers are also included below.

During the review process there was some discussion of novelty regarding the contributions of this work. As outlined previously, and also in our responses to Reviewer 2 below, we are firmly of the opinion that this paper contains important novelty and valuable research outcomes, as well as providing clear indications for necessary future work and a solid foundation on which that work might be built.

If anything here or in our responses requires further clarification, we'll be happy to provide it.

Edward Hart
(corresponding author)

Wind turbine main-bearing lubrication - Part 2: Simulation based results for a double-row spherical roller main-bearing in a 1.5 MW wind turbine

**Updated response to reviewer 1 (post revisions)**

Dear reviewer,

Thank you for taking the time to review our manuscript, and for your helpful comments which we have used to improve the quality of this paper. We include your comments below in **blue**, followed by our updated responses in **black**, having now made the promised edits.

The manuscript illustrates the application of EHD concepts explained in Part 1 to a practical case of a spherical roller bearing in a main shaft of a wind turbine. Properties of a commercial grease for this application are used as input parameters and the calculation includes starvation effects variable contact load plus potential dynamic effects. The manuscript shows that the contacts experience mixed-lubrication regime in a large proportion of the running time.

The reviewer is favourable to the publication of the manuscript but requires some minor revisions or at least the answer of some questions.

Revisions:

1. Section 3.1: It is very interesting to see the source of the load and speed cases analysed in this study. The authors refer to the IEC design standards. Can the authors point out exactly which standard they use (number?).

Yes indeed, we are referring to IEC 61400-1:2019 "Wind energy generation systems - Part 1: Design requirements". We have now referenced the exact standard in the updated manuscript.

2. Section 3.2: Perhaps the only missing aspect is to say that the considered properties of the grease are taken when the grease is fresh. It is well known that these properties change as the grease ages, but also the bleeding rate, so starvation will depend on the age of the grease somehow. The reviewer assumes that the authors did not consider re-greasing intervals in their model.

Grease ageing and re-greasing intervals were indeed not accounted for in this work, we have added a statement to this effect in Section 3.2 to make this point clear. The added text reads "Effects related to grease ageing and re-greasing (Lugt, 2013) were not considered in this analysis."

3. Section 3.2.1: What about cold starts or cold weather in Wind Turbines?

Excellent point, we have now included this in the final discussion section of Part 2. The added text reads "Finally, future work should also consider possible effects from intermittent wind turbine operation and cold starts on lubrication."

4. Section 4.2: For an EHL person $\Lambda$ values are a good way to sense the mean lubrication conditions in a contact. However, for bearings the ISO standards use $\kappa$ (defined in ISO 281). In the same standard an approximation between the two parameters is given $\kappa \approx \Lambda 1.3$, therefore it is possible to give order of magnitude of the results also in terms of $\kappa$. Notice that if this parameter is known, bearing life estimations are possible (also using ISO 281) and at least relative life values of different lubrication conditions can be obtained.

This is an excellent point and we agree that the link to $\kappa$ (defined in ISO 281) is relevant to include in the paper to give the reader a clear picture for how these things link together. We have therefore added the following text into the Background section (Section 2) of the manuscript "Finally, it is relevant to mention that in rolling bearing design standards (ISO, 2007) lubrication conditions are considered, but via the viscosity ratio (κ) rather than the film parameter (Λ) directly. The two quantities are linked by the approximate relationship κ ≈ Λ^1.3 (ISO, 2007). Since the aims of the present paper are to analyse lubrication conditions, film thickness values and related effects for the modelled main-bearing, this link is not pursued further in the current work."

5. Section 4.2: Indeed the consideration of starvation in the present manuscript is somehow a bit disappointing. However, the reviewer understands that the modelling of starvation in greased lubricated bearings is not simple. Especially when the availability of grease in the contacts is unknown and also the aging status of the grease (bleeding rate). Besides all these in this application the bearings should (in general) be fully packed of grease, which means that there is also a "gravity" effect. The grease in the lowest part of the bearing might have to carry the weight of all other grease making the bleeding non-uniform.

Yes, it would have been nice to provide a more detailed assessment of starvation. Unfortunately, this is not possible at the current stage, even when considering more recent work on starvation which has now been added to the "Part 1" review. We are hoping that this work serves as motivation for more detailed analyses with the kinds of sophisticated models that could try and capture such things, or alternatively laboratory experiments – some of which we are planning on performing ourselves.

6. Section 4.3: Grease thickener interactions, perhaps a more accurate calculation could be done with models: Nogi, (https://doi.org/10.1080/10402004.2020.1778147) and Morales-Espejel (Tribology International 74 (2014) 7–19). Indeed, a worked grease will see reduced these benefits, but how much?

Agreed, and we're hoping that we might manage such modelling in the future. But, unfortunately it is not something we were able to undertake in the current work. However, please note that our consideration of grease thickener effects does allow for some consideration for whether these effects will be important here, with results indicating that they won't based on what's available in the literature and our predicted film thickness values.

Section 4.3: Perhaps a more significant effects are stand-still periods and accelerations + decelerations. Sudden changes of speed. Stormy weather.

Yes these types of events could well be important here. As indicated above, the need for future work to consider effects of intermittent operation has now been included as part of the final discussion.

Section 5: The authors have written the following text "Rolling bearing fatigue life predictions are made under the assumption that fully EHL conditions hold throughout the bearing lifetime". In the experience of this reviewer this is inaccurate. Actually the parameter $\kappa$ takes into account the lubrication conditions as part of the factor $aISO$ in the standard ISO 281. Modern bearing life models can model variable operating conditions within the life of a bearing, not only speed but also load. Besides this, new bearing life models are being developed to explicitly separate surface and subsurface failure modes, the authors can look for such references (the reviewer has seen this at least in some SKF publications).

Thank you again for pointing out this oversight on our part. We have now provided information on the link between film thickness and the design standards (see related response above) to tie things together for the reader. Since life estimation brings with it a host of additional models, assumptions and caveats, performing a more detailed analysis linking with the approaches outlined in the design standards falls out of scope of the aims of the current paper (which is seeking to focus on an analysis of lubrication conditions, film thickness values and related effects). Furthermore, proper inclusion of such additional material would result in too long a paper. Therefore, we have indicated the link to ISO 281 in Section 2, but removed further discussions later in the paper, including our incorrect assertions you highlighted in Section 5.

Wind turbine main-bearing lubrication - Part 2: Simulation based results for a double-row spherical roller main-bearing in a 1.5 MW wind turbine

**Updated response to reviewer 2 (post revisions)**

Dear reviewer,

Thank you for taking the time to review our manuscript, and for your helpful comments which we have used to improve the quality of this paper. We include your comments below in **blue**, followed by our updated responses in **black**, having now made the promised edits.

The submitted paper discuss the lubricating conditions of main-bearings of wind turbines using a double-row spherical roller bearing in a 1.5 MW wind turbine as case study. In the reviewer's opinion the submitted paper is well written and structured. However, the scientific significance of the applied approach is limited. Understandably the authors apply known methods and a significant number of assumptions, however, this leads to the question of how the proposed approach differs from the typical industrial approach.

We provided a detailed response to this comment prior to revising the paper. That full response is still included below (in grey) for info. In short, we still believe that this paper offers important and identifiable novelty, since, such an analysis for the main-bearing in a wind turbine does not currently exist in the scientific literature. Furthermore, we are also able to shed light on the possibilities of more complex effects such as grease interactions and non-steady effects for this wind turbine component.

In response to this point, and some others below, it is important to discuss the aims and position of this paper in the context of the literature for main-bearings in wind turbines. The main-bearing is a component which has been identified as a problem for the wind industry, but one for which the literature has been relatively sparse until recently. A growing number of studies have now considered various aspects of the operation and reliability of this components, but, before now there have been no papers whatsoever which consider lubrication and EHL conditions for this component. Since the main-bearing is a tribological and lubricated component this aspect of its operation must be included if the premature failures seen in the field are to be properly understood and ultimately prevented. The difficulty when it comes to understanding this component is the fundamentally multi-disciplinary nature of the problem, since it is a bearing affixed to a flexible structure whose input loads are complex and determined by interactions between the wind turbine rotor and turbulent wind fields. Bringing these various aspects of the problem together necessitates simpler analyses to begin with, in order for important aspects and interactions to be identified – allowing areas for further analysis (using more sophisticated models) to be identified and prioritised. This paper represents that first effort for main-bearings in wind turbines. As such, we don't necessarily claim that we are doing things very much differently from the current industrial approach, but, the methods we use are applied transparently and with careful consideration of their validity and limitations etc. Furthermore, the industrial work on main-bearings you refer to isn't available in the public domain and so can't be built on as ours can. Finally, as with the review portion of the paper, a principal aim here is also the promotion of interdisciplinary understanding for engineers in related disciplines to be able to perform similar analyses and understand how our results impacts them. The current paper also achieves this goal and is the first to do this (for the main-bearing), this is another aspect of the paper's novelty, in addition to the new knowledge being presented here for this component. Overall we believe that this paper presents a solid foundational

set of results, using existing methods applied with careful consideration, which lays the groundwork for further work and more sophisticated analyses.

The authors focus on reaching a wider readership by using simplified descriptions and methods. While this approach seems reasonable for part 1, it leads to the major short-coming of part 2: apparent missing novelty. In the reviewer's opinion the submitted paper could be shortened significantly and complemented with further aspects such as: consideration of sliding (due to the formation of a load zone, due to low loads and along the contact line - Heathcote slip), in-depth analysis of temperature distribution, lubrication type, geometry (such as modern tapered main-bearings), starvation….

Again, a detailed response was provided previously and this is included again below.

As described above we believe these is important novelty in this paper already. In future work we are going to be working on some of the things you mention here, but they fall outside the aims and context of this work. For example, wind turbines never see steady operating conditions, as such analysis of slip requires a fully dynamic main-bearing model. Such a model will require significant development beyond our current capabilities and so falls out of scope for this work at present. Similarly, an in-depth analysis of the temperature distribution would require thermal modelling of the main-bearing is not yet available in our model. With respect to temperature we know that the temperature range we study is close to the values seen in practise and, importantly, the results indicate that these temperatures may be seen as a change point between lubrication regimes. This result is valuable and important in of itself and indicates where future analysis should be focussed. As stated above, we believe the results presented provide an important baseline on which to build. We agree that the analyses you suggest should be undertaken, but we believe they fall into the category of 'recommended future work'.

In the reviewer's opinion the submitted paper is a case study rather than a research paper.
Since this paper focusses on an individual wind turbine and a specific main-bearing it is perhaps fair enough to consider it a 'case study'. I don't agree that this means it is not also a 'research article'. As outlined above, this is currently the only article in the literature which has presented results of this nature for a wind turbine main-bearing, meaning it provides important research novelty despite focussing on a specific turbine and bearing.

Please consider following points:
Abstract
• The temperature should be introduced as an influencing factor.
This has been added to the abstract as requested.
Introduction and Background
• The authors state that
      o "higher than expected failure rates" and "premature failures in main-bearings" occur. Please quantify and give the necessary references.
This has been done as requested (see the added footnote and references on the first page).
      o "it cannot currently be known a priori whether main-bearing failure rates are likely to be improved or worse". In the reviewer's opinion the authors could discuss whether the contact conditions would most likely positively or negatively impact the lubricating conditions (speed, load, time between contacts and temperature could be compared).
Having considered this question we have come to the conclusion that we are actually not currently able to answer this, and prefer not to speculate since things could go either way! This is because the main-bearing geometry and number of rollers will change with wind turbine size. While the loads

will increase and the rotational speed is likely to decrease (for reason of aerodynamic efficiency), the increased diameter and number of rollers may offset these changes. Therefore, at this stage we believe that it remains correct to say we don't yet know how things will change for larger turbines, both with respect to lubrication conditions and failure modes.

> o "fatigue life assessment……explicitly assumes EHL". Modern lifetime calculations do not assume EHL. For example, the aiso or the askf consider the lubricating conditions through the viscosity ratio Kappa.

Thank you for pointing out this error, we have added the following text in Section 2 to provide this information "Finally, it is relevant to mention that in rolling bearing design standards (ISO, 2007) lubrication conditions are considered, but via the viscosity ratio (κ) rather than the film parameter (Λ) directly. The two quantities are linked by the approximate relationship κ ≈ Λ^1.3 (ISO, 2007). Since the aims of the present paper are to analyse lubrication conditions, film thickness values and related effects for the modelled main-bearing, this link is not pursued further in the current work.".

> o "the described load structures were found to drive large, rapid variations..". Please quantify.

This has been done as requested.

> • Based on the investigations in Guo et al. (2021) the authors state that axial motions are slow and highly unlikely to impact the lubricating film. However, they state that "the lubrication was not modelled directly". In this case, how was the aforementioned conclusion drawn?

This has been clarified in the updated manuscript. The added text reads "Note, lubrication was not modelled directly in this previous study[2]  -> (footnote) ->  [2] Instead, axial velocities of individual bearing rollers were estimated from measurements and compared to the mean entrainment velocity, ũ. Literature results on axial velocity influences were then leveraged in order to conclude that the observed axial motions would not be expected to disturb the lubricant film."

> • Sentences 29 through 38 are a good example of strongly simplified explanations which could be strongly shortened

We have carefully read through the paper and edited where we felt the language could be shortened or improved.

> • Please give the coordinates of the two radial components of the load vector

Apologies, but we're not sure what you mean by 'coordinates' here as there is no reference frame or coordinate system being presented.

> Methodology
> • Please use references where is appropriate (e.g. IEC design standards)

This has been done as requested.

> • Could the authors please state if "10 min." refer to simulation or operation time?

It refers to operational time. This has now been clarified in the manuscript.

> • The authors explain that the applied force was considered (instead of the reaction force). Furthermore, they explain that it is necessary to ensure the correct one is calculated with the respect to the reference frame. How do the results will deviate if the reaction force is used instead? They have equal magnitude

This is important for tracking the loads on individual rollers as they move around the main-bearing. The Hertzian contact model takes an input of the applied load vector, with the vector direction determining the location of applied load etc. The rollers are simultaneously orbiting the bearing centre and so their positions relative to applied loading matters. If the reaction load is used the load will be of the correct magnitude, but acting in the wrong direction and so maximally loading the wrong rollers. We have reworked some of the description around this in the manuscript to try and ensure clarity.

> • Please comment why the upwind row is "only occasionally loaded during normal operations". One would expect that the radial load is carried by both rows

A wind turbine sees very large thrust loads, this causes axial deflections which tend to unload the upwind row throughout much of the operating time of the turbine. The second row is there mostly for when the turbine if not operating. This is shown in https://doi.org/10.1002/we.2549 and also provides motivation for the design changes proposed in https://doi.org/10.1007/s10010-021-00462-1 . Both references are now given at the relevant point in the method description (step 4).

• In the reviewer's opinion the authors should expand their consideration of roller sliding. It is well known that sliding have a great impact on the contact temperature and the film formation

As mentioned above, a wind turbine never sees steady-state operating conditions and so standard sliding models cannot be used to evaluate this. A full dynamic model would be needed instead. This is something we intend to develop but not a capability we have at this time. Furthermore, it is our understanding that sliding will indeed affect the bearing temperature through friction, but that its effect on film thickness is relatively small because it is the viscosity (and hence temperature) of the lubricant at the contact inlet that most directly determines film height. So, the main effect of sliding would be to elevate the overall temperature of the main-bearing. But, this is something we already account for indirectly because the temperature range we use is based on measured temperatures of operating main-bearing casings. We will seek to consider sliding in future work, but we are of the opinion that the current paper presents important findings and novel research outcomes in its current form.

• The authors comment that "other details are proprietary". In the reviewer's opinion profile and roughness measurements can be carried out and consider in half-space calculations

We have checked and are currently unable to provide information in more detail that it is currently presented.

• Table 1
    o The combine surface-roughness seems high. Please give the individual values and the method used for determining the roughness

We are not at liberty to provide individual values, but these values have been double checked for correctness. The values are combined by taking the square-root of the sum of squares of the values.
    o Please comment on the profile in width-direction

As a spherical roller the profile is provided by the curvature radius used to calculate reduced radii in x and y direction. Note, the Ry value in which this information is combined is now included in the manuscript (footnote 4 on page 11).

o "Inlet temperature" is commonly used for "lubrication nozzles". Please consider writing "contact inlet"

Excellent suggestion, thank you! This has been done as requested throughout the manuscript.

• In the reviewer's opinion the selected temperatures could be too low. If the outer bearing casing is at 20-40 °C degrees the contact temperature (most likely) won't be in the range between 30 - 40°C. How did the authors make this assumption?

These are low speed bearings and so, based on information we could find on casing versus internal temperatures, we don't expect the internal temp to be much higher than that of the casing. In addition, and as discussed in the paper, the values are chosen because they are certainly within the normal operating range for a wind turbine main-bearing (based on available data) but also because they represent a transition point between lubrication regimes (as described in the manuscript). Higher temperatures are possible, but this would only strengthen the conclusion of the paper that mixed lubrication is expected. The above points are discussed in Section 3.2.1 and it is also indicated that "… Lubricant contact-inlet temperatures may therefore be higher than the values listed above"

• Sentences 177 through 185 are a further example of strongly simplified explanations which could be strongly shortened. In which cases would the authors consider a spherical-roller bearing as a point contact (aside from insignificant loads)?

The definition of point contact, e.g. see "Part 1" paper, includes both circular and elliptical contacts under any load level. Similarly, line contact refers to all contacts resulting in a rectangular contact patch. Therefore, yes we consider and SRB roller to be a point contact in the context of Hertzian theory. However, as shown in the paper, we argue that it should be treated as a line contact for the purposes of EHL film analysis. To clarify further, a point contact is one which initially contacts at a point, and a line contact initially contacts along a line. In both cases the contact then becomes something else as soon as load is applied. This is all standard regarding definitions in Hertzian theory.

• One could argue that the consideration of starvation is not scientifically accurate. In the reviewer's opinion the authors should either select a concrete approach or take the sub-chapter out. If the authors decide to expand this chapter please consider modern literature regarding starvation and the influence of base oil and thickener on the film formation.

We agree that this analysis of starvation is limited. But on reviewing the most recent available literature on starvation (as suggested by yourself in "Part 1" – where the starvation section is now much expanded), it remains the case that current analytical formulas are not yet at the stage where they may be practically applied to a full bearing. However, we believe that the analysis as presented still provides useful information by considering the impacts of film reductions of the general order of magnitude seen during starvation. This helps show that starvation is a key consideration for this main-bearing since it could be the different between full EHL and mixed lubrication. However, we agree that the language and context of this sub-section of the paper needed improving. We have therefore edited the text to make it clear that starvation as treated in this paper is a crude order of magnitude estimate of possible starvation effects only. We believe it is important to still include these results, but they are now presented with the proper caveats in place.

Results

• Sentences 231 through 240 are a further example of strongly simplified explanations which could be strongly shortened.

As in a previous comment, we have reviewed all language in the paper and improved for clarity and length where we felt it necessary.

• Please consider using the shaft speed instead of the wind speed for Figure 10.

I assume you are referring to Fig 2. Wind speed is used as this is the variable that dictates the operating point of a wind turbine. The rotational speed of the wind turbine only changes for part of it's operating range and so would result in a less clear figure. Furthermore, the 'speed' from a lubrication point of view is already represented as part of the dimensionless viscosity/speed parameter L. For these reasons we believe wind speed is the better candidate for the color-scale in this plot.

• In the discussion of Figure 10 the authors state that around the bearing circumference an unloaded region is generally included. While this is true for radial loads the authors stated before, that the down-wind bearing was used as case study. This bearing is supposed to be axially loaded, which could result in a load zone over 360°.

Yes this is indeed correct, and 360 degree loading can happen, but it is more common that not that there will be an unloaded region on the bearing. This comment is there to explain why a full range of M (dimensionless load) values is seen across all wind speeds – and the abundance of M values of the left shows that unloaded rollers are common.

• The authors explain that "outlying points at very low loads correspond to high values of film thickness". Please mark this points int Figure 10

On reflection we have decided there is no real gains from adding to the figure, but we did feel the discussion on this point could be improved. Therefore, we have updated the description around this point to make things clearer for the reader.

• Please revise sentence 254-255. It gives the impression that the load is the determining influencing factor

Yes you're right that this part of the text needed improving, we have now updated this part of the discussion and added a footnote to clarify that the influence of load here is only due to its value going to zero. We highlight that in general other parameters have a more dominant impact on film thickness.

- It is commonly known that the pressure increases in the area of the PETRUSEVICH-peak. How do the authors assume that the maximum EHL pressure values are equivalent to those seen in dry contacts?

We use the Hertzian contact pressure as an estimate of maximum pressure in the conjunction, which is the case for heavily loaded and or slow moving rollers (we discuss this in "Part 1" of the paper). In the manuscript we do indicate that this is an estimate of the true maximum, and to ensure this is clear we have added further comments on this in Section 4.1 ("Contact pressure results indicate that, with respect to the effects modelled currently and assuming maximum pressures are well approximated by the Hertzian value, bearing material ultimate-strength limits are not being exceeded during operation.") and Section 5. Note also that experimental results often tend to show that the petrusevich-peak is in reality much smaller than that predicted by numerical models, with the global maximum well estimated by the maximum Hertzian value in dry contact. E.g. see *Albahrani, S., Philippon, D., Vergne, P., and Bluet, J.: A review of in situ methodologies for studying elastohydrodynamic lubrication, Proceedings of the Institution of Mechanical Engineers, Part J: Journal of Engineering Tribology, 230, 86–110, 2016.*

- Did the authors consider the axial load in the pressure calculations?

Yes, pressure values were calculated after the Hertzian contact model was used to evaluate all rollers loads under both radial and axial loads applied to the bearing.

- Please comment whether the minimal loading of the bearing is guaranteed and how this affect its sliding behaviour

For an SRB in a wind turbine, minimal loading is not guaranteed as there is no preloading. As has been shown in previous work https://doi.org/10.1002/we.2549 unloading of the main-bearing can occur during operation. This would likely introduce sliding into the system but we are not yet able to quantify this effect.

- Please give the used formula for lambda

This has been added as requested.

- Please comment if the selected variations step in table 2 are plausible. Furthermore, the magnitude of pressure influence is dependent on the load magnitude (as shown in Fig 3.), therefore, the simplified statement that by reducing the load by 50 % the lambda value increases by 6 % is misleading.

Yes these are plausible changes in variable values based on the data available to us. In each case this has been indicated earlier in the manuscript for temperature changes seen for main-bearing data, available data on possible values of alpha, and in the case of load with respect to load variations obtained via different levels of modelling. With respect to the pressure/load comment I believe you are referring to the fact that the change in film thickness with change in load is different at different loads. This is certainly true, although the same is also true for the other varied parameters, e.g. temperature, when they are changed. But, please note that the values given in the sensitivity results table are average values of changes in Lambda from sensitivity testing across the whole dataset of operating points. Therefore, this summarises the average influence at around different operating points. The key outcome from the load part of the analysis is to show insensitivity to loads being halved in general. Importantly this indicates that more sophisticated models of loads would be unlikely to arrive at different conclusions to those we make here.

- Why did the authors consider the bearing as a point contact for chapter 4.3?

Again we should clarify that our definition of point contact includes both elliptical and circular contacts. Here we are treating it as a long elliptical point contact. For lubrication it was treated as a line contact for reasons that were outlined. But in 4.3 we need to understand changes in its dimensions as an elliptical contact (changes in both a and b) since this is what it is. The reasons that

we needed to use equivalent line contact equations before do not apply here, and so it was deemed appropriate and necessary to perform an elliptical contact analysis. There is no contradiction between these approaches, since one considers contact patch dimensions while the other relates to selecting an appropriate film thickness analysis approach.

• Why was a threshold of 25 % selected? Please add references

We have added references to show where this value was obtained. Further discussion on this is also included in the 'non-steady effects' section of Part 1 of the study, also referenced at this point in the manuscript.

Discussion and conclusions

• The authors should discuss the influence of running-in

On reflection we feel that running-in is beyond the scope of discussion in the present work since it would require a fairly detailed outline of what it is and why it might be important. While we agree it will be influential, we feel that the already listed effects which should be considered in the future (skidding, skew, intermittent operation, detailed analysis of starvation, detailed modelling of grease interactions) likely take precedence in terms of the immediate next steps in this research area.

• No further comments. Please consider the points above

---

## Author Response (AR2)

**Wind turbine main-bearing lubrication - Part 2: Simulation based results for a double-row spherical roller main-bearing in a 1.5 MW wind turbine**

Dear Amir,

Thank you again for handling the review process for this submission. Please see below the comments provided and a summary of how we have incorporated them into the updated manuscript.

**Comment (reviewer):** Spherical roller bearings, unlike tapered roller bearings, are not preloaded. Often times, the roller elements within these bearings experience skidding when passing through the unloaded zone. It would be beneficial to inform the reader about the implications of this possible skidding in the film thickness.

Response: We agree that this is a point worth making in the paper. We have therefore added a discussion on the likely presence of some amount of individual roller slip outside of the loaded zone. This addition reads as follows. "The lubricant entrainment velocity is calculated assuming pure rolling, see Appendix B. Recent work by Bergua et al. (2021), in which up-tower measurements from an operational wind turbine were found to show no appreciable levels of main-bearing gross-slip, provides some justification for this assumption. In reality, some amount of individual roller slip would be expected to occur outside of the loaded zone, even in cases of no gross-slip. Such effects are not accounted for in this analysis. However, the lack of gross-slip observed for an operational main-bearing (Bergua et al., 2021) indicates that the pure rolling assumption is likely valid throughout most of the loaded zone and, importantly, at the point of maximum load". In the discussion section we then elaborate further on the possible implications of slip for lubrication and film thickness values "It will also be necessary to determine the impact of effects not accounted for here, including housing and bedplate flexibility and dynamic roller behaviour (sliding, skewing etc.). The presence of individual roller sliding/slip, through the unloaded zone, could result in high levels of friction as the roller re-enters the loaded zone and is accelerated back to pure rolling. While slip isn't expected to strongly influence film thickness values directly (Crook, 1961), increases in frictional energy may influence the temperature of the bulk lubricant in the main-bearing. Where a roller is sliding, non-Newtonian lubricant properties, such as shear thinning, may also become more important (Bair, 2005)."

**Comment (editor):** Thank you for addressing the comments. It would be nice to discuss a bit about the uncertainties associated with the results and elaborate more on the main bearing lubrication trend in larger turbines in the discussion

**Response:** Thank you for this suggestion. We would like to highlight that some of this is already covered in the discussion section, for example, we emphasise the main areas of uncertainties "As has been emphasised, uncertainties are present which mean that care must be taken when interpreting these findings. In particular, further work is needed to better understand properties related to key sensitivities, these being temperature, starvation and $\alpha*$ values, for wind turbine main-bearings and their lubricating greases". This, combined with the sensitivity analysis presented in Table 2, we feel provides the reader with a good understanding of where the main uncertainties are, and the effect on film thickness results of these parameters having different values from those used. However, we do agree that it is useful to discuss whether these results might be extrapolated to larger wind turbines. Therefore, the following paragraph has been added to the discussion "At this stage, it is not clear how the results of this analysis might change for larger wind turbines. Indeed, it is difficult even to speculate. While larger wind turbines will experience increased load magnitudes

and generally rotate more slowly, the main-bearing will likely be of greater diameter and contain more rollers. The former will increase roller loads and decrease entrainment velocities, while the latter tend to have opposite effects. The overall drivetrain design, including the main-bearing(s), may also be quite different for larger wind turbines. Lubrication conditions in larger turbines will therefore ultimately be determined by the interplay of these various factors."